# CSB promoter downregulation via histone H3 hypoacetylation is an early determinant of replicative senescence

Clément Crochemore[1,2], Cristina Fernández-Molina[1,2,3], Benjamin Montagne[1,2], Audrey Salles[4] & Miria Ricchetti [1,2]*

Cellular senescence has causative links with ageing and age-related diseases, however, it remains unclear if progeroid factors cause senescence in normal cells. Here, we show that depletion of CSB, a protein mutated in progeroid Cockayne syndrome (CS), is the earliest known trigger of p21-dependent replicative senescence. CSB depletion promotes overexpression of the HTRA3 protease resulting in mitochondrial impairments, which are causally linked to CS pathological phenotypes. The CSB promoter is downregulated by histone H3 hypoacetylation during DNA damage-response. Mechanistically, CSB binds to the p21 promoter thereby downregulating its transcription and blocking replicative senescence in a p53-independent manner. This activity of CSB is independent of its role in the repair of UV-induced DNA damage. HTRA3 accumulation and senescence are partially rescued upon reduction of oxidative/nitrosative stress. These findings establish a CSB/p21 axis that acts as a barrier to replicative senescence, and link a progeroid factor with the process of regular ageing in human.

[1] Institut Pasteur, Stem Cells and Development, Department of Developmental and Stem Cell Biology, 75015 Paris, France. [2] CNRS UMR 3738, Team Stability of Nuclear and Mitochondrial DNA, 75015 Paris, France. [3] Sorbonne Universités, UPMC, University of Paris 06, IFD-ED 515 Paris, France. [4] Institut Pasteur, UTechS Photonic BioImaging PBI (Imagopole), Centre de Recherche et de Ressources Technologiques C2RT, Paris, France. *email: miria.ricchetti@pasteur.fr

Senescence, which is a process that limits proliferation of damaged cells in response to various types of stress, has been associated with organismal ageing[1]. Accumulation of senescent cells in tissues, as well as withdrawal of senescent stem cells from the regeneration process, are marks of tissue deterioration linked to ageing[2] and may contribute to organ degeneration and aged-related diseases. Senescence also acts as a barrier to cancer[3] and plays a physiological role in normal development and tissue homeostasis[4–6].

Senescence is characterized by a proliferative arrest of metabolically active cells[7], which is distinct from other forms of cell cycle arrest (e.g., quiescence) by its essentially irreversible nature, and by acquisition of specific features, including increased cell size, β-galactosidase activity, persistent DNA damage-response (DDR), presence of heterochromatin foci, and highly active secretory activity (senescence-associated secretory phenotype, or SASP)[8]. Senescence triggered by replication-induced DNA damage, or telomere shortening that determine a persistent DDR, and resulting in the stabilization of the transcription factor p53 and expression of the cyclin-dependent kinase inhibitor p21 (coded by $p21^{Waf1}$)[9,10], is called replicative senescence. Stimuli such as genomic and epigenomic damage, oxidative and proteotoxic stress, and mitogenic signals that engage either the DDR-dependent (p53/$p21^{Waf1}$) pathway or activation of the CDKN2 locus through expression of the tumor suppressor p16 (encoded by $p16^{Ink4}$) also induce senescence[11]. High levels of p16 are observed late during replicative senescence following the p21 trigger, probably to maintain the senescent state[12].

Senescence allows tissue repair and remodeling through the SASP[11]. Moreover, induced senescence promotes cell reprogramming in various tissues through paracrine factors[13,14]. Conversely, exhaustion of the stem cell pool results from paracrine induction of senescence in non-damaged cells[15]. Furthermore, defective clearance of senescent cells, which is particularly relevant during ageing, induces a persistent and deleterious SASP[16]. In this context, specific elimination of $p16^{Ink4}$-positive cells in a progeroid mouse model delays the onset, or limits the progression, of aged-related symptoms[17].

Premature cellular senescence has been described in mesenchymal stromal cells differentiated from Werner syndrome-induced pluripotent stem cells (iPSCs)[18], and in other progeroid syndromes, such as the Hutchinson-Gilford progeria syndrome[19] and Ataxia-telangiectasia[20]. Similarly, human primary fibroblasts depleted of BLM, WRN, or RECQ4 helicases, which are responsible for precocious ageing in the Bloom syndrome, Werner syndrome, and Rothmund–Thomson syndrome, respectively, display multiple markers of senescence compared to normal fibroblasts. Moreover, keratinocytes from progeroid Cockayne syndrome, mutated in CSA and with impaired DNA repair capacity, display early senescence[21]. Although suspected of being implicated in normal ageing, defects observed in progeroid syndromes have not been formally demonstrated to drive physiological ageing.

These findings raise the possibility that factors responsible for Cockayne syndrome (CS), namely CSA and CSB, are effectors of cellular senescence, and also question which activity of these multifunctional proteins is determinant. In this context, we identified a novel stress-induced protease activity that improperly degrades functional mitochondrial proteins in cells from CS patients[22]. Notably, in CS cells combined oxidative and nitrosative stress (ROS and RNS) promote overexpression of the HTRA3 protease, which results in degradation of the mitochondrial DNA polymerase POLG1 responsible for replication of the organelle genome, thereby triggering mitochondrial dysfunction. The cognate HTRA2 protease is also overexpressed in CS cells, but it is unclear whether POLG1 degradation is promoted by HTRA3 alone or in combination with HTRA2. Although the mechanism leading to HTRA3 (and HTRA2) overexpression in CS cells remains undefined, it is likely uncoupled from the well documented CSA/CSB involvement in the repair of ultraviolet (UV)-damaged DNA through transcription-coupled nucleotide excision repair (TC-NER)[23]. Indeed, these proteins are also transcription factors and have been implicated in chromatin remodeling in vitro (in particular CSB)[24]. CSA/CSB have also been detected in mitochondria where they repair oxidative damage of mitochondrial DNA[25]. Critically, scavenging of oxidative and nitrosative species with a porphyrine derivative, MnTBAP, led to rescue of the cellular and molecular defects in progeroid CS patient cells[22].

Here, we report that the CS alterations of HTRA3, HTRA2, POLG1 and mitochondria are recapitulated during replicative senescence of human primary fibroblasts. We also show that these alterations and replicative senescence are triggered by downregulation of CSB, through reduced histone acetylation of its promoter. Consequent CSB depletion results in the exposure of the $p21^{Waf1}$ promoter to activation, which leads to senescence, and this activity of CSB is independent of its function in UV-induced DNA repair.

## Results

**HTRA3 overexpression during replicative senescence**. To assess whether HTRA3, which is considered a prevalently mitochondrial protease[26], was expressed during cellular senescence, we examined population doubling of three independent IMR-90 serially passaged human embryonic fibroblasts (Fig. 1a). Cells at passage numbers (PN) indicated with an arrow were selected for in-depth investigation, and are representative of distinct phases: proliferative PN16, PN19, PN23; the end of exponential growth, PN27; pre-senescent PN31; and senescent PN35. Senescence-associated beta-galactosidase staining (SA-β-gal, Fig. 1b and Supplementary Fig. 1a), as well as increased cell size (Supplementary Fig. 1b, c), confirmed pre-senescence at PN31 and senescence at PN35.

Interestingly, at PN31 and PN35 HTRA3 immunofluorescence (IF) was twofold higher compared to earlier passages, independent of increased cell size (Fig. 1c and Supplementary Fig. 1b, d). Super-resolution structured illumination microscopy (SIM) of early-passage and senescent fibroblasts revealed that HTRA3 is not prevalently mitochondrial (several mitochondria do not display HTRA3 and a large part of the labeling remains extra-mitochondrial), and this condition does not appear to change upon senescence (Fig. 1d). Increased HTRA3 levels during replicative senescence were confirmed by Western blot (WB), in particular the short isoform at PN31, and cleaved products of the long isoform at PN35 (Fig. 1e and Supplementary Fig. 1e). During stress, long-HTRA3 undergoes autocleavage that generates shorter L-forms[27]. Real-time quantitative PCR (RT-qPCR) analysis confirmed high levels of HTRA3 transcripts, in particular the long form, in senescent cells at PN35, together with the established senescence markers $p21^{Waf1}$, $p16^{Ink4}$, and the SASP marker IL-6 (Fig. 1f). The levels of HTRA3 (short) and $p21^{Waf1}$ transcripts were 1.5- and twofold higher, respectively, also in pre-senescent PN31 cells compared to earlier passages. Increased levels of HTRA3 were not dependent on declined cell proliferation, since slow dividing/non-dividing early-passage fibroblasts at confluence, assessed by decline of the cell cycle markers cyclin A2 and PCNA, did not display higher levels of HTRA3 (RNA and protein) compared to cells undergoing robust proliferation (Supplementary Fig. 2a–c). Absence of senescence in the above-mentioned cells was verified by unaltered levels of p21 and $p21^{Waf1}$ as well as p16 and $p16^{Ink4}$. Thus, HTRA3 is expressed at high levels during replicative senescence, and as early as established senescence markers, in particular the principal effector $p21^{Waf1}$.

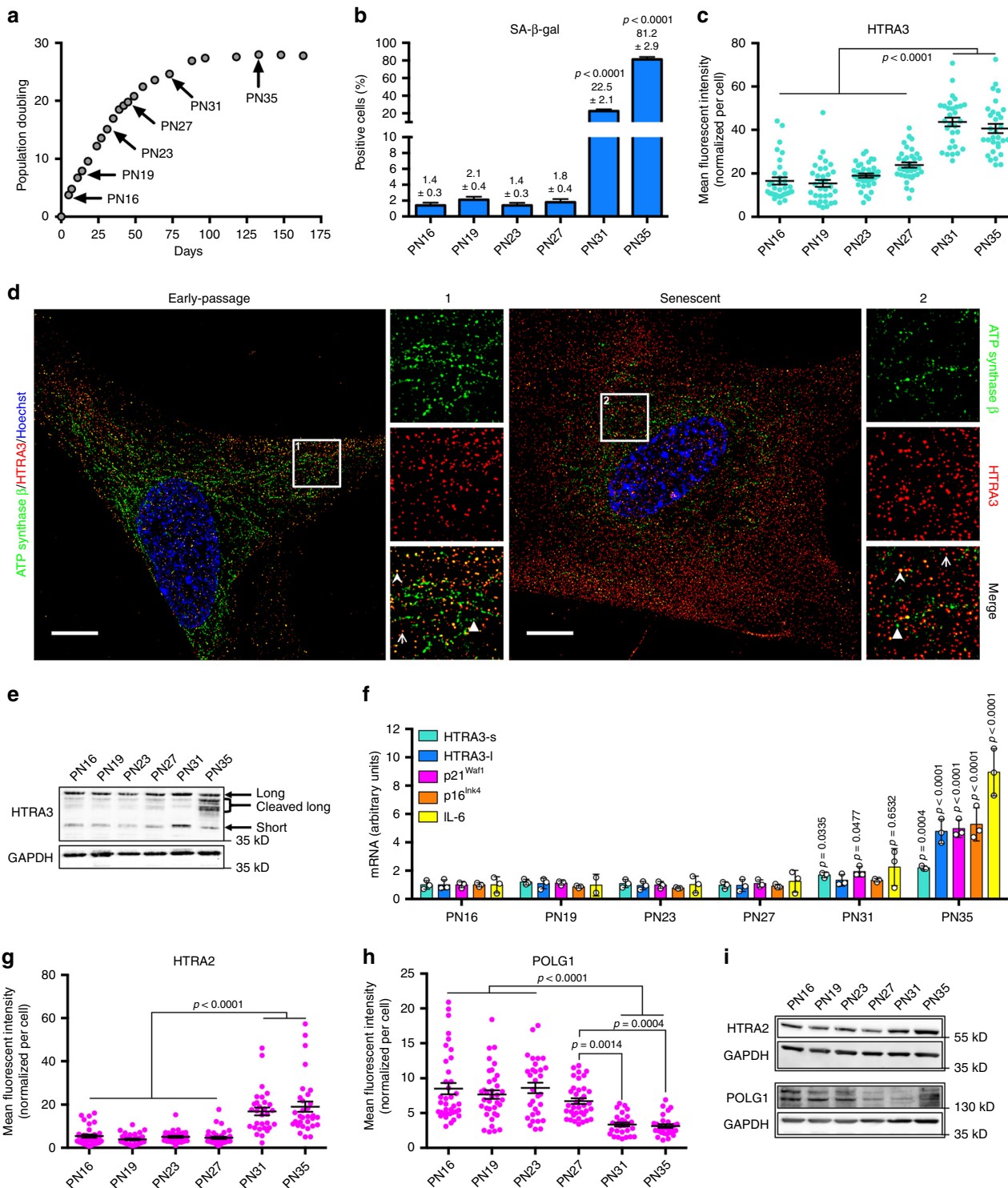

In CS cells, also the HTRA2 protease was overexpressed[22] and this was also the case for HTRA2 transcripts (Supplementary Fig. 3a) and protein (Fig. 1g, i and Supplementary Fig. 3c) in senescent cells (PN35).

**Mitochondrial impairment during replicative senescence.** In cells from CS patients, accumulation of HTRA3 (alone or in combination with HTRA2) was associated with depletion of the mitochondrial DNA polymerase POLG1, despite high levels of *POLG1* transcripts, suggesting degradation of this polymerase[22].

Accordingly, we observed reduced levels of POLG1 by IF (Fig. 1h and Supplementary Fig. 3d) and WB (Fig. 1i) in pre-senescent (PN31) and senescent (PN35) cells, despite unchanged or increased levels of transcripts (Supplementary Fig. 3b). Cells kept at confluence for 1-2 days displayed slightly increased levels of HTRA2 and reduced levels of POLG1 (Supplementary Fig 2a–c), suggesting that these proteins are to some extent dependent on factors other than replicative senescence. In CS cells, POLG1 depletion was associated with increased ROS and reduced mitochondrial ATP production[22]. Senescence (Supplementary Fig. 4a–d) was

**Fig. 1** Overexpression of HTRA3 and mitochondrial impairment in replicative senescence. **a** Cumulative population doubling of IMR-90 fibroblasts (starting from PN15). Senescence corresponds to plateau (proliferative arrest). Cells analyzed at PNs identified with black arrows; $n = 3$ independent cultures; mean ± SD, values are reported in the Source Data file. **b** Percent of SA-β-gal⁺ cells; $n = 1100$–1340 cells (PN16-PN27) and $n = 180$–410 cells (PN31-PN35) from three independent experiments, mean ± SEM (values indicated on the top of columns); one-way ANOVA ($F = 768.9$, DFn = 5, DFd = 5551, $p < 0.0001$) with post-hoc Tukey's test vs. PN16. **c** Quantification of mean HTRA3 fluorescence intensity per cell at indicated PN (representative images in Supplementary Fig. 1d). **d** Sublocalisation of HTRA3 and mitochondria in early-passage (PN16) and senescent (PN35) IMR-90 fibroblasts with SIM (one plan of a z-stack acquisition for each cell). Cells immunolabelled for ATP synthase β (green) to reveal mitochondria, and HTRA3 (red); nuclei counterstained with Hoechst (blue). Scale bars = 10 μm. For each cell, a x2.5 magnification of a region (1 or 2) is shown on the right with immunostaining for ATP synthase β, HTRA3, and merge (representative arrowhead for HTRA3/ATP synthase β colocalization, arrow for extra-mitochondrial HTRA3, and triangle for mitochondria with no HTRA3 signal detection). **e** Immunoblot of HTRA3, and GAPDH (loading control, reprobing after stripping). α-HTRA3 antibody recognizes long and short isoforms (Supplementary Fig 1e). **f** Quantitative RT-qPCR of *HTRA3* (*short* and *long* form), *p21*$^{Waf1}$, *p16*$^{Ink4}$, and *IL-6* transcripts. $n = 3$ independent experiments; mean ± SD; one-way ANOVA (*HTRA3s*: $F = 13.12$, DFn = 5, DFd = 12, $p = 0.0002$; *HTRA3l*: $F = 29.12$, DFn = 5, DFd = 12, $p < 0.0001$; *p21*$^{Waf1}$: $F = 63.24$, DFn = 5, DFd = 12, $p < 0.0001$; *p16*$^{Ink4}$: $F = 38.71$, DFn = 5, DFd = 12, $p < 0.0001$; *IL-6*: $F = 29.75$, DFn = 5, DFd = 12, $p < 0.0001$) with post-hoc Tukey's test vs. PN16 ($p$-values on the top of scatter plots/columns). Quantification of mean fluorescence intensity (mFI) of **g** HTRA2 and **h** POLG1 immunolabeling per cell (representative images in Supplementary Fig. 3c, d). **i** WB of POLG1, HTRA2, each with the loading control GAPDH. IFs: $n = 30$–50 cells from three independent experiments; mean ± SEM; one-way ANOVA (**c** $F = 58.32$, DFn = 5, DFd = 194, $p < 0.0001$; **g** $F = 30.75$, DFn = 5, DFd = 194, $p < 0.0001$; **h** $F = 16,67$, DFn = 5, DFd = 193, $p < 0.0001$) with post-hoc Tuckey's test. IF measurements include normalization to cell size. Source data are provided as Source Data files.

associated with increased levels of oxidative stress, measured by reduced glutathione (GSH), a strong scavenger of ROS, and its ratio with oxidized glutathione (GSSG)[28] (Supplementary Fig. 4e), and to some extent mitochondrial ROS (Supplementary Fig. 4f, g). Senescent cells displayed reduced ATP production by mitochondrial oxidative phosphorylation (OXPHOS), and decreased levels of mitochondrial complexes I, III, and IV, which were also reduced during pre-senescence (Supplementary Fig. 4h, i). Thus, senescent cells recapitulate cellular and mitochondrial alterations observed in CS patient cells.

**CSB depletion is an early event in replicative senescence**. We then asked whether altered HTRA3 and POLG1 levels during replicative senescence were a consequence of CSB impairment, since CSB mutation resulted in these defects in CS cells. We observed a progressive and dramatic decrease of *CSB* transcripts from PN27 to PN35 (from twofold to eightfold, respectively, Fig. 2a), confirmed by WB at the end of the exponential phase (PN27) (Fig. 2b), and by IF in pre-senescent and senescent fibroblasts (Fig. 2c, d). CSB depletion was not observed in slowly dividing/non-dividing early passages fibroblasts (Supplementary Fig. 2a–c). Thus, reduced expression of CSB was detected earlier than the appearance of senescence, preceding the established senescence marker *p21*$^{Waf1}$ and SA-β-gal⁺ staining, as well as accumulation of HTRA3/HTRA2, and depletion of POLG1. Interestingly, transcripts of *CSA*, the other factor mutated in CS, did not decline during senescence (Fig. 2a), whereas the protein was reduced by less than 50% in pre-senescent (PN31) and senescent (PN35) cells, therefore at a lower extent and later PN than CSB (Fig. 2b).

**CSB knockdown induces premature p21-dependent senescence**. To verify whether CSB depletion triggers cellular senescence, we knocked down CSB in early-passage IMR-90 fibroblasts with two shRNA (shCSB#1, shCSB#2, Fig. 3a–c). Cells transduced with control shRNA (shSCR) progressively decreased *CSB* transcripts prior to increased expression of *p21*$^{Waf1}$ and *HTRA3* (black columns in Supplementary Fig. 5a–c), thereby confirming downregulation of *CSB* during senescence. The direct correlation observed between *p21*$^{Waf1}$ and *HTRA3* (Fig. 3d), supports the notion that increased expression of both transcripts is strictly linked in senescence.

*CSB* downregulation, which was maintained until PN20 and PN21 for shCSB#1 and shCSB#2, respectively, was nevertheless sufficient to promote cellular senescence. Figure 3e shows proliferation arrest of *CSB*-knocked down fibroblasts earlier (PN26, eight passages post-transduction of shCSB#1; PN24, after six passages for shCSB#2) compared to cells transduced with shSCR (PN28, ten passages post-transduction). Precocious senescence upon transient *CSB* knockdown was confirmed by a higher number of SA-β-gal⁺ cells at each passage, and to a larger extent for the more efficient shCSB#2 (Fig. 3f and Supplementary Fig. 5d), with an initial burst as early as PN19 and PN20. Further, the levels of *HTRA3* and *p21*$^{Waf1}$ in *CSB*-silenced fibroblasts were higher than control beyond PN19-PN20 (Supplementary Fig. 5b, c).

We then focused on PN19 and PN20, when *CSB* was efficiently knocked down, and observed a 15-fold increase of *p21*$^{Waf1}$ mRNA as early as PN19 with shCSB#2 compared to shSCR, and a fivefold increase at PN20 (Fig. 3g). Higher levels of p21 with both shRNAs at PN20 were confirmed by WB (Fig. 3i). Moreover, both shCSBs induced a significant increase of HTRA3 transcript and protein at PN20 (Fig. 3h, i). WB also revealed an increase of HTRA2 and a decrease of POLG1 levels with both shCSBs (Fig. 3i), confirming our results (see Fig. 1g–i), and consistent with the alterations observed in CS cells. Interestingly, cumulated data from shSCR and the two shCSBs (Supplementary Fig. 5a, b) demonstrate a strong inverse correlation between *CSB* and *p21*$^{Waf1}$ expression (Fig. 3j). Taken together, these data indicate that low CSB levels trigger p21-dependent replicative senescence.

We wondered whether CSB affects p21 expression by direct interaction with its promoter. Chromatin immunoprecipitation (ChIP) assays showed that CSB directly interacts with two regions of the *p21*$^{Waf1}$ promoter that are target of p53[29], and a more extended interaction in the distal (5') region was observed with the C-terminal anti-CSB than with N-terminal anti-CSB (Fig. 4a–c). As a control, both antibodies failed to display a signal at the TATA box that is close to the transcription start site. Moreover, cells stably expressing the eGFP reporter under control of the *CDKN1A* (*p21Waf1*) promoter (Fig. 4d) expressed higher levels of eGFP upon CSB silencing (shCSB#2), assessed with IF (Fig. 4e, f) and WB (Fig. 4g), further showing that CSB directly regulates p21 expression.

Premature cellular senescence upon CSB silencing did not result in increased DNA damage at PN19 and PN20 assessed by immunofluorescence of the double-strand break (DSB) repair factors γ-H2AX and 53BP1[30] (Supplementary Fig. 5e–h). Immunoblots of γ-H2AX confirmed this result and showed no increased phosphorylation of the DNA damage signaling factor ATM (pATM), (Supplementary Fig. 5i). These data suggest that

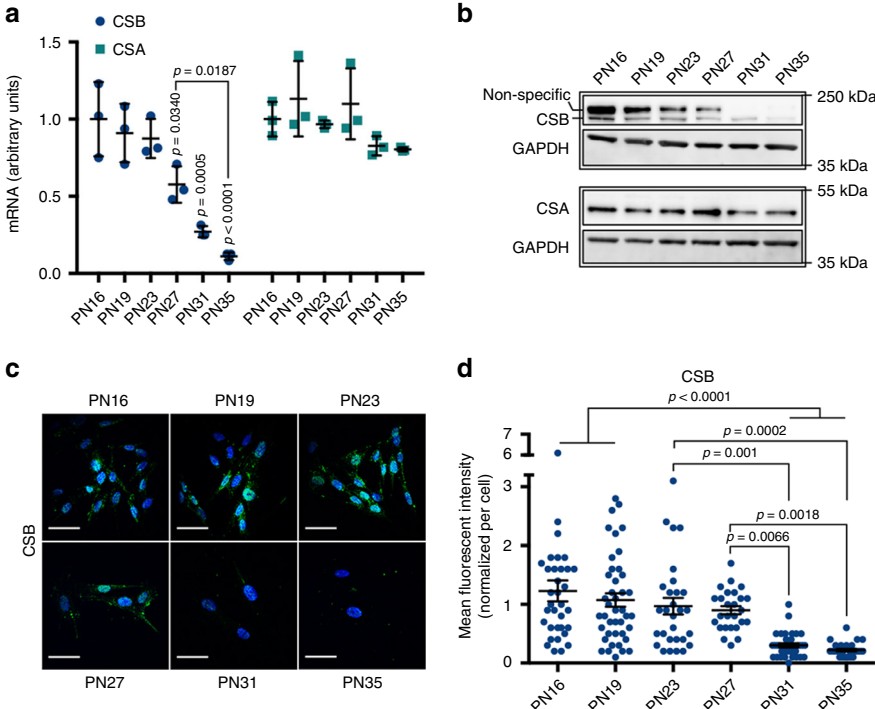

**Fig. 2** CSB depletion is an early event in replicative senescence. **a** RT-qPCR of *CSB* and *CSA*. $n = 3$ independent experiments; mean ± SD; one-way ANOVA (*CSB*: $F = 19.40$, DFn = 5, DFd = 12, $p < 0.0001$; *CSA*: $F = 2.55$, DFn = 5, DFd = 12, $p = 0.855$) with post-hoc Tukey's test vs. PN16 (*p*-values on the top of scatter plots/columns) when not specifically indicated. **b** WB of CSB and CSA, each with the loading control GAPDH (lower blot, GAPDH F-c staining). Upper band in CSB blot is non-specific[67]. **c** Representative confocal acquisitions of cells immunostained for CSB (green) and counterstained with Hoechst (blue) after maximum intensity projection with Imaris, scale bar = 50 μM, and **d** quantification of the CSB mFI/cell. $n = 30–50$ cells from three independent experiments; mean ± SEM; one-way ANOVA ($F = 13.76$, DFn = 5, DFd = 189, $p < 0.0001$) with post-hoc Tukey's test vs. PN16 when not specifically indicated. IF measurements include normalization to cell size. Source data are provided as Source Data files.

CSB depletion is a primary event leading to senescence, and not a secondary effect of DNA damage-induced senescence.

Finally, we observed that ectopic overexpression of CSB in a fraction of early-passage IMR-90 fibroblasts resulted in lower p21 expression (Supplementary Fig. 6a, b) and a smaller percentage of SA-β-gal+ cells (Supplementary Fig. 6c, d) compared to cells carrying an empty vector, suggesting that senescence was delayed upon increased CSB levels.

**CSB depletion by histone H3 hypoacetylation of its promoter**. We then investigated the mechanism responsible for CSB downregulation during replicative senescence. UV-induced CSB repression has been recently described in age-related nuclear cataract through coordinated DNA hypermethylation and histone deacetylation at the CSB promoter[31]. Histone hypoacetylation is associated with compact chromatin and repressed transcription[32]. RT-qPCR of DNA methyltransferases *DNMT3a* and *DNMT3b*, which are responsible for de novo DNA methylation, did not reveal differences among the various cell passages (Fig. 5a). However transcripts of *DNMT1*, the enzyme responsible for DNA methylation maintenance during replication, progressively decreased at PN31 and PN35, probably due to reduced replication in pre-senescent and senescent cell populations, as suggested by lower levels in poorly dividing early-passage fibroblasts (Supplementary Fig. 2c). We then examined the methylation profile of the CSB promoter (positions in Supplementary Fig. 7) by pyrosequencing after bisulphite treatment. At all passages, we observed low methylation levels for CpG(#8), which was hypermethylated in age-related nuclear cataract, as well as the other seven CpG sites in the promoter region of CSB (Fig. 5b). Thus, methylation alterations do not occur in this region during

replicative senescence. However, 24 h treatment with 10 μM, but not 2 μM, of the methyltransferase inhibitor 5-aza-2′-deoxycytidine (5-aza-dC) (Supplementary Fig. 8a, b) resulted in enhanced transcription of *CSB* (Supplementary Fig. 8c) and protein content (Supplementary Fig. 8d), indicating that global DNA hypomethylation is associated with CSB overexpression.

Conversely, the levels of acetylated histone H3 decreased at the end of the exponential growth (PN27), and to a larger extent in pre-senescent (PN31) and senescent (PN35) cells (Fig. 5c). The decrease in histone H3 acetylation largely exceeded the global depletion of H3, and this was not dependent on cell cycle arrest since levels of acetylated H3 histone were not altered in slowly dividing/non-dividing IMR-90 (Supplementary Fig. 2a). Accordingly, mRNA levels of histone deacetylase HDAC1 (and of highly homologous HDAC2) increased transiently at PN27 (Fig. 5d), and were reduced at later passages. Remarkably, the peak of HDAC1 expression and histone H3 hypoacetylation appeared at the switch point for CSB depletion, i.e., PN27 (see Fig. 2), suggesting a role of this histone modification in the regulation of CSB expression during replicative senescence. ChIP on the CSB promoter (positions in Supplementary Fig. 7) showed reduced levels of acetylated histone H3 at PN27 and PN31 compared to the previous passages (Fig. 5e), whereas the levels of global histone H3 were not altered. Thus, hypoacetylation of histone H3 at the CSB promoter directly correlates with CSB downregulation during replicative senescence.

Furthermore, treatment of early-passage fibroblasts (PN16) for 24 h with increasing concentrations of anacardic acid (AA), an histone acetyl transferases (HAT) inhibitor (Fig. 5f), resulted in reduced CSB transcript and protein, concomitantly with a global reduction of acetylated histone H3 (Fig. 5g, h). A specific

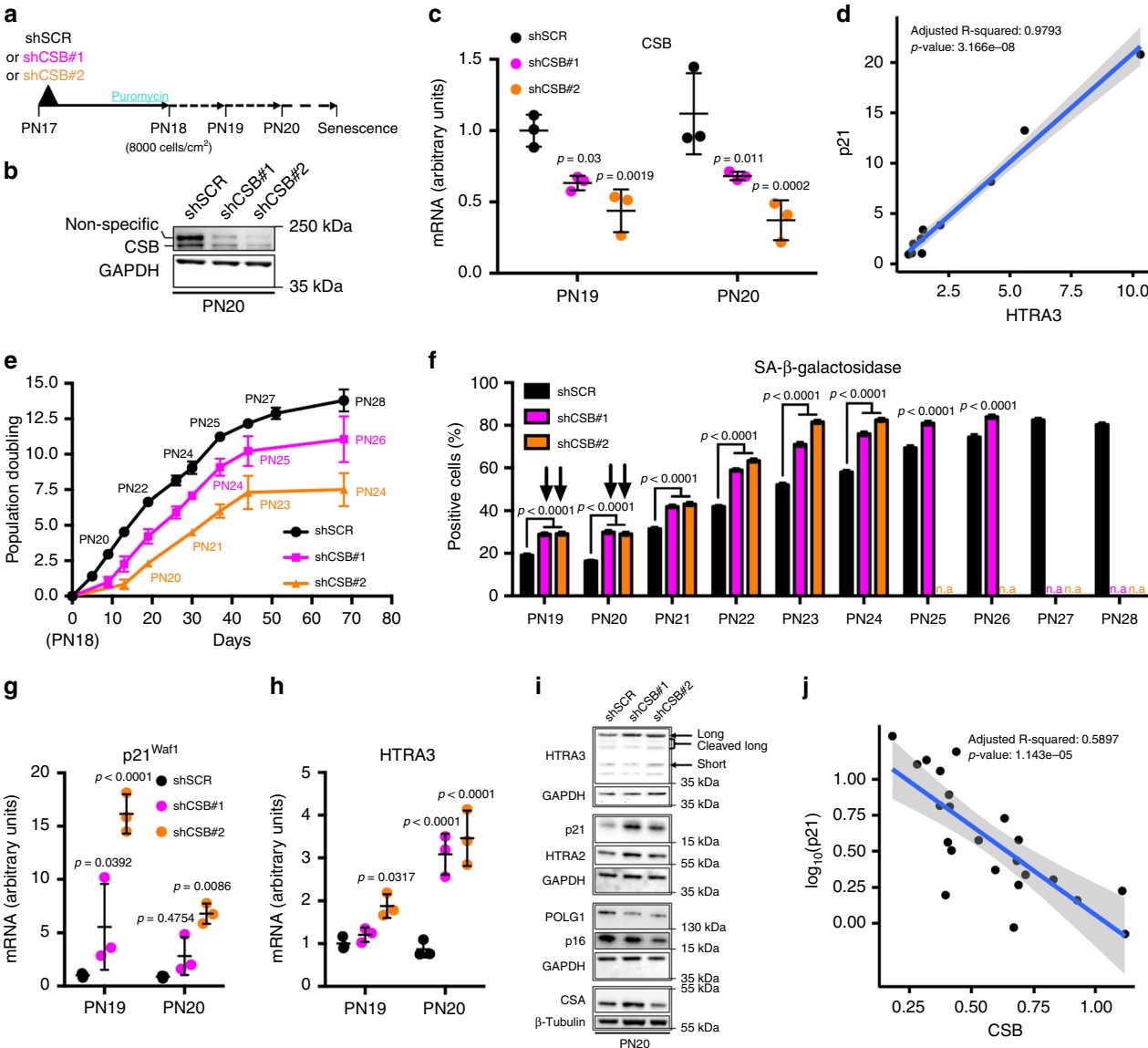

**Fig. 3** CSB knockdown induces premature p21-dependent senescence. **a** Scheme of the experiment. **b** Immunoblot of CSB at PN20. GAPDH was used as a loading control. **c** Quantitative RT-qPCR of *CSB* at PN19 and PN20 in IMR-90 fibroblasts knocked down for *CSB* (shCSB#1 and shCSB#2) and scramble control (shSCR). $n = 3$ independent experiments, mean ± SD, values are reported in the Source Data files; two-way ANOVA ($F = 28.24$, DFn = 2, DFd = 12, $p < 0.0001$) with post-hoc Tukey's test vs. shSCR. **d** Direct correlation between *p21^Waf1* and *HTRA3* transcript in control shSCR (from data in Supplementary Fig. 5b, c). **e** Cumulative population doubling of serially passaged IMR-90 (starting at PN18, $n = 3$ independent cultures). **f** Quantification of SA-β-gal⁺ cells from PN19 to PN28 (cultures stopped growing at PN26 with shCSB#1 and at PN24 with shCSB#2); $n = 1160$-4480 cells/condition from three independent experiments, mean ± SEM; two-way ANOVA (PN19-24: $F = 542.3$, DFn = 2, DFd = 40677, $p < 0.0001$. PN25-26: $F = 78.71$, DFn = 1, DFd = 5114, $p < 0.0001$) with post-hoc Tukey's (PN19-24) or Sidak's (PN25-26) tests vs. the respective shSCR; n.a = not applicable. Arrows indicate the initial burst of SA-β-gal⁺ staining upon CSB silencing. SA-β-gal⁺ cells in shSCR at PN19-PN20 probably result from response to lentiviral infection[68]. RT-qPCR of **g** *p21^Waf1* and **h** *HTRA3* at PN19 and PN20. $n = 3$ independent experiments, mean ± SD; two-way ANOVA (*p21^Waf1*: $F = 44.26$, DFn = 2, DFd = 12, $p < 0.0001$. *HTRA3*: $F = 35.04$, DFn = 2, DFd = 12, $p < 0.0001$) with post-hoc Tukey's test vs. shSCR. Data in panels **c, g, h** (which are limited to PN19 and PN20) are extracted from panels in Supplementary Fig. 5a–c. **i** WB of HTRA3, p21, HTRA2, POLG1, p16, and CSA at PN20. Samples on the same blot are framed; each frame displays the respective GAPDH or β-tubulin used as a loading control (middle blot, GAPDH F-C staining). High levels of p16 in shSCR are compatible with lentiviral infection[69]. **j** Linear regression of each individual value in Supplementary Fig. 5a (*CSB*, x-axis) vs. log10-transformed values in Supplementary Fig. 5b (*p21^Waf1*, y-axis). Source data are provided as Source Data files.

reduction of H3 acetylation of the CSB promoter was then observed by ChIP upon AA treatment (Fig. 5i). Interestingly, at high AA concentrations HTRA3 and p21 levels increased after further 24 h in culture and upon drug withdrawal (Fig. 5h), consistently with the late upregulation of these senescence markers following CSB depletion (see above, Fig. 1c, e, f). Under these conditions, H3 acetylation and CSB levels were restored.

CSB depletion upon reduced global H3 acetylation was also obtained with a variety of inhibitors that target the main HATs (MG149, CPTH2, C646, and curcumin), Supplementary Fig. 8e. Moreover, at least C646 and CPTH2 displayed H3 hypoacetylation at the CSB promoter, assessed by ChIP (Supplementary Fig. 8g). Conversely, the histone deacetylase inhibitor MS275 that increased H3 acetylation at the global

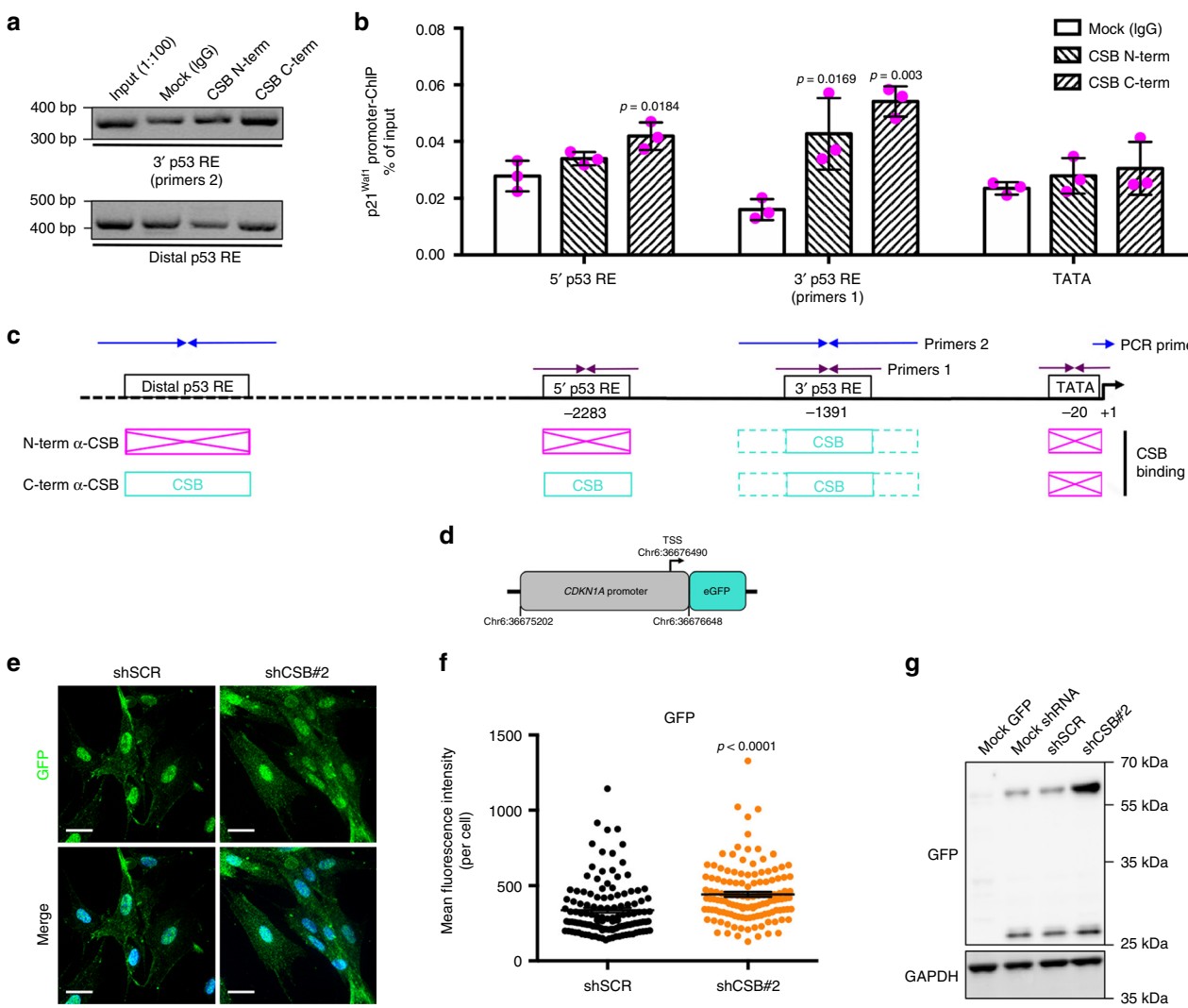

**Fig. 4** CSB binds to the *p21^Waf1* promoter. **a** PCR (agarose gel) and **b** quantitative PCR (histogram) analyses of several DNA fragments in the *p21^Waf1* promoter from ChIP assays with either N-term α-CSB [Bethyl] or C-term α-CSB [Abcam] antibodies. **c** Scheme not at scale of primer positions on the *p21^Waf1* promoter and summary of positive amplifications in turquoise (boxes in magenta indicate no amplification). $n = 3$ independent experiments; one-way ANOVA (5'p53RE: $F = 7.698$, DFn = 2, DFd = 6, $p = 0.0220$. 3'p53RE: $F = 17.11$, DFn = 2, DFd = 6, $p = 0.0033$; TATA: $F = 0.8699$, DFn = 2, DFd = 6, $p = 0.4659$) with post-hoc Tukey's test for each tested regions vs. Mock. **d** Scheme of the *CDKN1A* (*p21^waf1*) promoter-eGFP reporter plasmid. **e** Representative confocal acquisitions of cells stably expressing the *CDKN1A* (*p21^waf1*) promoter-eGFP reporter upon shCSB#2 silencing or control shSCR, immunostained for GFP (green) to increase the signal of endogenous GFP, and counterstained with Hoechst (blue) after maximum intensity projection with Imaris; scale bar = 50 μM, and **f** quantification of the GFP mFI/cell; $n = 127$ cells/condition from three independent experiments, mean ± SEM; unpaired *t*-test (two-tailed) ($t = 4.555$, DF = 253), *p*-value vs. shSCR. **g** WB of GFP and GAPDH (loading control) in cells not expressing (Mock GFP) or stably expressing (Mock shRNA) the *CDKN1A* (*p21^waf1*) promoter-eGFP reporter, not silenced (shSCR) or silenced for CSB (shCSB#2). Cells that express GFP display a band at 27 kDa, and also an upper band at around 60 kDa, in particular upon CSB silencing, corresponding to dimers observed in other conditions[70]. Source data are provided as Source Data files.

level (Supplementary Fig. 8f) and specifically at the CSB promoter (Supplementary Fig. 8h), did not alter CSB levels (Supplementary Fig. 8f), suggesting that global as well as local hyperacetylation of H3 does not increase CSB expression. Taken together, these experiments show that CSB depletion in replicative senescence is triggered by hypoacetylation of its promoter.

**CSB-dependent replicative senescence is independent of p53.** Increased levels of p21 during replicative senescence depend on p53 activation. p53 protein, but not transcript (except at PN35) levels increased towards senescence, in agreement with previous findings[33], suggesting stabilization of the protein (Fig. 6a, b). P53 silencing (Fig. 6c, d) alone had little or no effect on the

population doubling compared to control shSCR (Fig. 6e). Silencing of p53 followed by silencing of CSB reduced the population doubling as much as silencing of CSB alone, suggesting that CSB depletion triggers replicative senescence independently of p53. This result was confirmed by treating fibroblasts in culture with pifithrin-alpha (PTF-α), a p53 inhibitor that blocks p53-mediated p21 activation[34], in the presence and in the absence of CSB silencing (Fig. 6f). Indeed, treatment with PTF-α reduced the population doubling slightly compared to untreated cells (Fig. 6g), but did not overtly affect the fraction SA-β-gal+ cells (Fig. 6h, i) and *p21^Waf1* expression (Fig. 6j, k), whereas CSB silencing strongly affected these readouts. PTF-α slightly increased the population doubling, reduced the fraction of SA-β-gal+ cells and *p21^Waf1* expression (thus confirming the

 

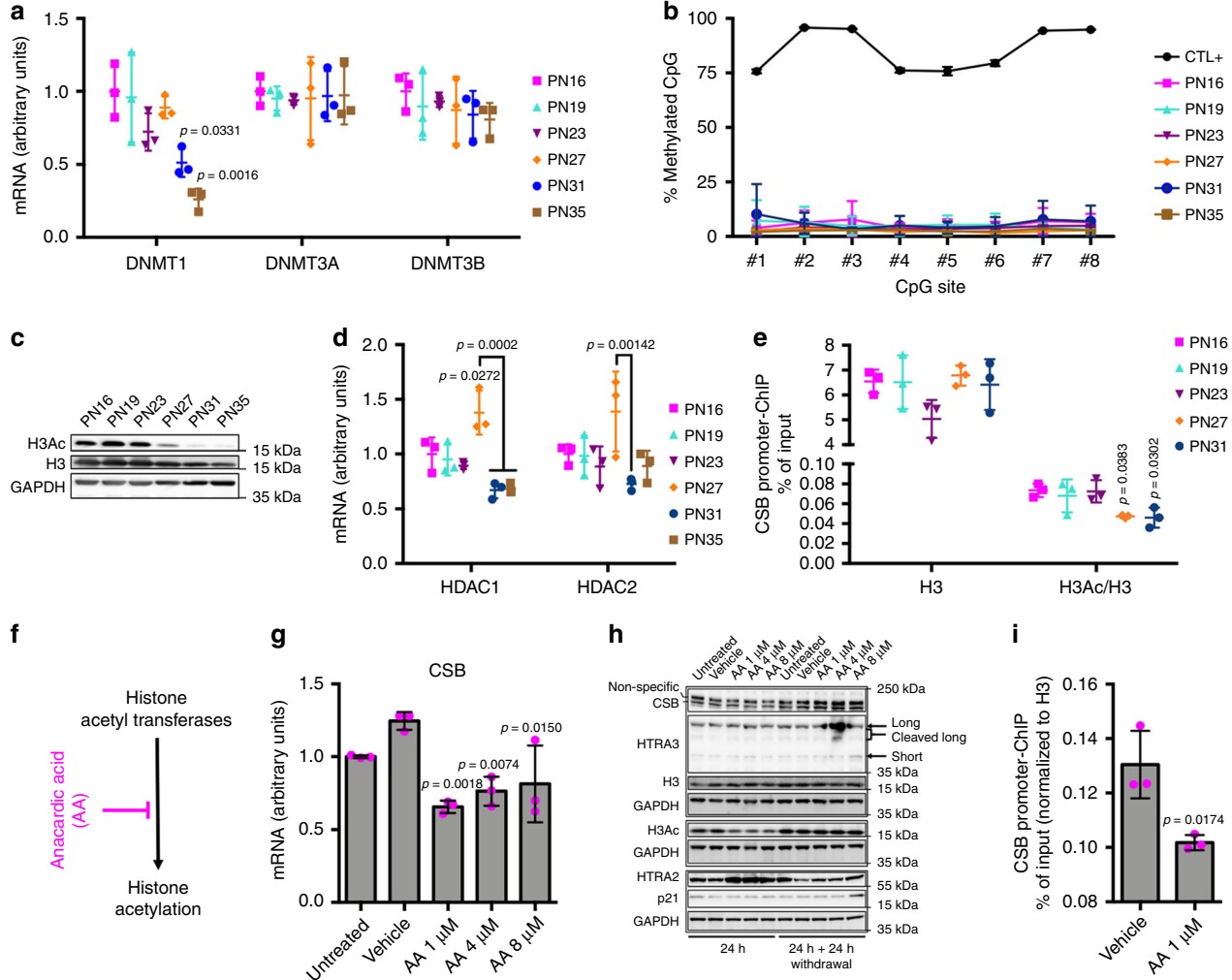

**Fig. 5** CSB depletion by histone H3 hypoacetylation of its promoter. **a** Quantitative RT-qPCR of *DNMT3A*, *DNMT3B*, and *DNMT1*. **b** DNA methylation (percent) of eight individual CpG sites in the *CSB* promoter at the indicated PN (positive control [CTL + ], a universal methylated (human) DNA standard). $n = 3$ independent experiments, mean ± SD; two-way ANOVA ($F = 2.983$, DFn = 5, DFd = 96, $p = 0.0151$) with post-hoc Tukey's test. **c** Immunoblots of H3 acetylated (H3Ac) and total histone H3 (reprobed after H3Ac stripping) at the indicated PN, with GAPDH as a loading control. **d** Quantitative RT-qPCR of *HDAC1* and *HDAC2*. **e** Quantitative PCR of DNA in a ChIP assay with either α-H3 acetylated or total α-H3 (material not available at PN35). Primers detect the occupancy of H3Ac/H3 at a specific region of the CSB promoter. Results are expressed as the percentage of input DNA normalized to H3 occupancy. **f** Scheme indicating the anacardic acid (AA) target. **g** Quantitative RT-qPCR of *CSB* upon treatment with increasing concentrations of AA or DMSO (Vehicle) for 24 h, and in untreated control (Untreated). **h** Immunoblot of H3 acetylated, total H3 histone, CSB, HTRA3, HTRA2, and p21 from whole-cell extracts with increasing concentrations of AA or DMSO (Vehicle) for 24 h, and 24 h after AA withdrawal and in the corresponding controls (Untreated). Samples on the same blot are framed; each frame displays the respective GAPDH used as a loading control (GAPDH (F-C staining in the middle blot)). Quantitative PCRs: $n = 3$ independent experiments; mean ± SD; one-way ANOVA (**a** *DNMT1*: $F = 9.129$, DFn = 5, DFd = 12, $p = 0.0009$; *DNMT3A*: $F = 0.05094$, Dfn = 5, DFd = 12, $p = 0.9980$; *DNMT3B*: $F = 0.5325$, DFn = 5, DFd = 12, $p = 0.7481$; **d** *HDAC1*: $F = 12.93$, DFn = 5, DFd = 12, $p = 0.0002$; *HDAC2*: $F = 3.826$, DFn = 5, DFd = 12, $p = 0.0264$; **e** H3: $F = 2.298$, DFn = 4, DFd = 10, $p = 0.1304$; H3Ac/H3: $F = 5.064$, DFn = 4, DFd = 10, $p = 0.0171$; **g** $F = 9.516$, DFn = 4, DFd = 10, $p = 0.0019$) with post-hoc Tukey's test vs. PN16 (or Vehicle, **g**) when not specifically indicated. **i** Quantitative PCR analysis of a DNA fragment in the CSB promoter from ChIP assay with α-H3 acetylated in the presence and in the absence of AA treatment; $n = 3$ independent experiments, mean ± SD; unpaired Student's $t$-test (two-tailed) ($t = 3.908$, DF = 4) vs. Vehicle. Source data are provided as Source Data files.

activity of the inhibitor) upon CSB silencing, thereby attenuating senescence, compared to CSB silencing alone. Thus, CSB-mediated replicative senescence is essentially independent of p53.

**CSB-dependent senescence is specific to the DDR/p21 pathway.** Replicative senescence of IMR-90 fibroblasts is correlated with activation of the *p21^Waf1* pathway (Fig. 1f and Supplementary Fig. 5b), which is DNA damage-response (DDR)-dependent[3]. To assess the extent of endogenous DNA damage during replicative senescence we enumerated γ−H2AX and 53BP1 foci at critical

PNs. Figure 7a and Supplementary Fig. 9a show a significant increase of both markers at PN27, concomitantly with CSB depletion (see above, Fig. 2), suggesting a link between these two events.

To verify that CSB downregulation and HTRA3 overexpression are specific to DDR-induced senescence, we triggered senescence by irradiating (10 Gy) early-passage IMR-90 fibroblasts (PN17) and examined cells after 10 days[35]. Notably, an increased number of γ−H2AX and 53BP1 foci in irradiated fibroblasts (Fig. 7b–d) correlated with a dramatic increase in SA-β-gal+ cells (89.2 ± 2.3% compared to non-irradiated cells 2.7 ± 1.1%; Fig. 7e).

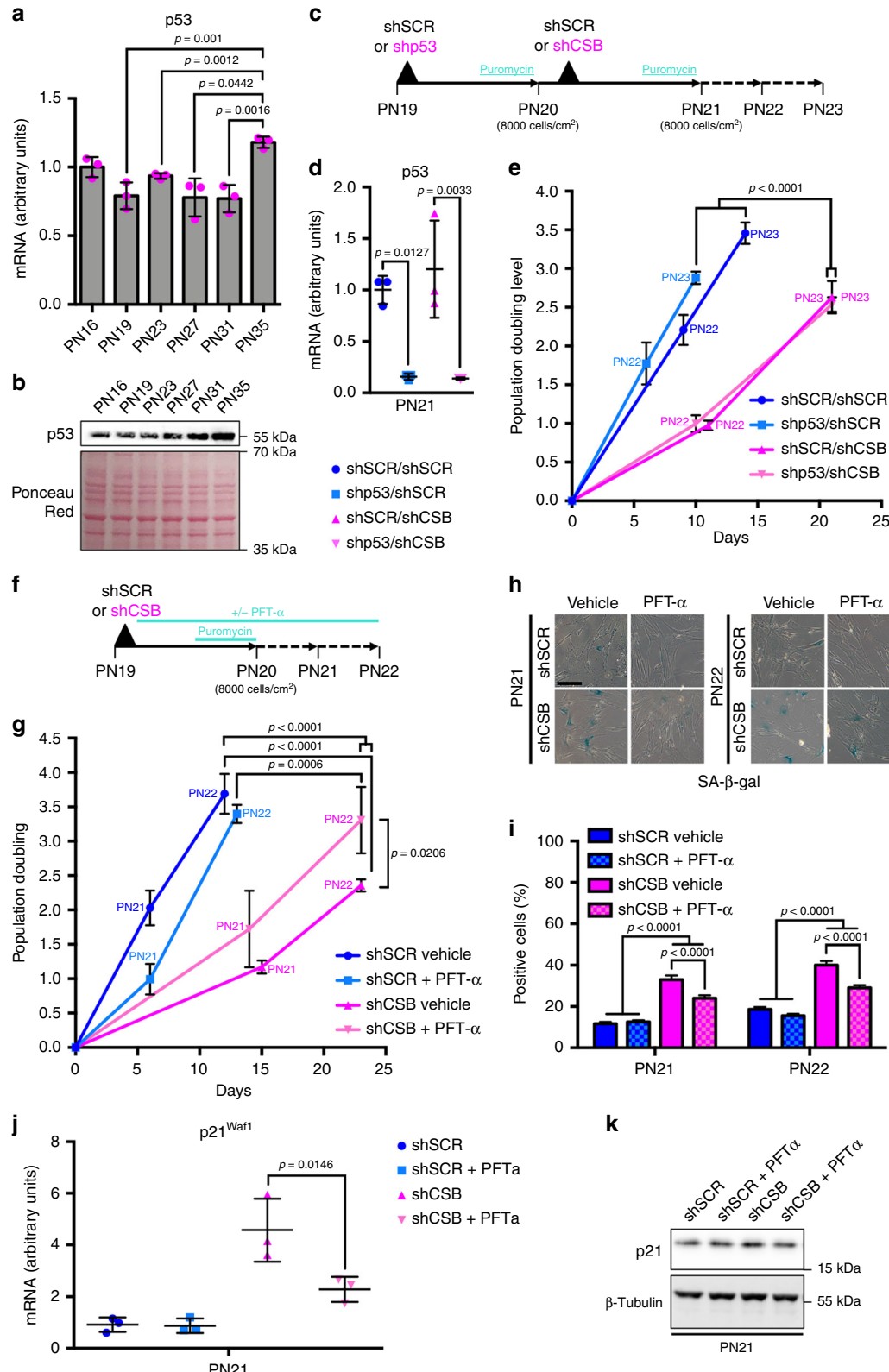

Senescence following irradiation was verified by high levels of the DDR-dependent senescence marker p21 (Fig. 7f). Conversely, no accumulation of the DDR-independent and senescence marker p16 was observed (Fig. 7f) including at earlier and later time points (7 days and 15 days post-IR, respectively, Supplementary Fig. 9b), differently from a study performed with cells at a later PN and displaying apparently lower basal levels of p16 before

irradiation than in our case[36]. Importantly, we observed decreased CSB, and increased HTRA3 and HTRA2 levels upon irradiation (Fig. 7f), as it was the case for replicative senescence (see Figs. 1 and 2). Decreased CSB levels upon irradiation resulted in the loss of CSB interaction with the $p21^{Waf1}$ promoter (ChIP assays, Supplementary Fig. 9c–f), supporting the notion that this interaction is a barrier to senescence. We note that POLG1 was

**Fig. 6** CSB-induced senescence is independent of p53. **a** RT-qPCR and **b** WB (Ponceau Red staining as a loading control) of p53 in IMR-90. **c** Experimental set-up of the two-step transduction of IMR-90, indicating also cell density at seeding. **d** RT-qPCR of p53 after silencing (PN21). RT-qPCRs: $n = 3$ independent experiments, mean ± SD; one-way ANOVA (**a** $F = 10.38$, Dfn = 5, DFd = 12, $p = 0.0005$; **d** $F = 15.37$, DFn = 3, DFd = 8, $p = 0.0011$) with post-hoc Tukey's test. **e** Population doubling after two passages in culture; $n = 3$ independent experiments, mean ± SD; GraphPad Prism two-by-two comparison of slope after linear regression (shSCR/shSCR vs. shp53/shSCR: $F = 8.87048$, DFn = 1, Dfd = 14, $p = 0.1$; shSCR/shCSB vs. shp53/shCSB: $F = 0.103672$, Dfn = 1, DFd = 14, $p = 0.7522$; shSCR/shSCR vs. shSCR/shCSB: $F = 80.735$, DFn = 1, Dfd = 14, $p < 0.0001$; shSCR/shSCR vs. shp53/shCSB: $F = 186,816$, Dfn = 1, DFd = 14, $p < 0.0001$; shp53/shSCR vs. shSCR/shCSB: $F = 79.2891$, DFn = 1, DFd = 14, $p < 0.0001$; shp53/shSCR vs. shp53/shCSB: $F = 163.975$, DFn = 1, DFd = 14, $p < 0.0001$). **f** Experimental set-up and **g** population doubling of IMR-90 transduced with either a non-targeted (shSCR) or a CSB-targeted (shCSB) shRNA and grown in presence ( PFT-α) or absence (Vehicle) of the p53 inhibitor pifithrin-α (10 μM). $n = 3$ independent experiments, mean ± SD; GraphPad Prism two-by-two comparison of slope after linear regression (shSCR Vehicle vs. shSCR + PFT-α: $F = 2.72661$, DFn = 1, DFd = 14, $p = 0.1209$; shCSB Vehicle vs. shCSB + PFT-α: $F = 6.67947$, DFn = 1, DFd = 14, $p = 0.0216$; shSCR Vehicle vs. shCSB Vehicle: $F = 157.235$, DFn = 1, DFd = 14, $p < 0.0001$; shSCR Vehicle vs. shCSB + PFT-α: $F = 40.3927$, DFn = 1, DFd = 14, $p < 0.0001$; shSCR + PFT-α vs. shCSB Vehicle: $F = 67.2499$, DFn = 1, DFd = 14, $p < 0.0001$; shSCR + PFT-α vs. shCSB + PFT-α: $F = 19.5273$, DFn = 1, DFd = 14, $p = 0.0006$). **h** Representative images and **i** quantification of SA-β-gal+ cells at PN21 and PN22. $n = 840$–1560 cells/condition from three independent experiments, mean ± SEM; for each PN one-way ANOVA (PN21: $F = 63.99$; DFn = 3, DFd = 4580, $p < 0.0001$; PN22: $F = 64.65$, DFn = 3, DFd = 4372, $p < 0.0001$) with post-hoc Tukey's test. **j** RT-qPCR of $p21^{Waf1}$; $n = 3$ independent experiments, mean ± SD; one-way ANOVA ($F = 19.29$, DFn = 3, DFd = 8, $p = 0.0005$) with post-hoc Tukey's test. **k** WB of p21; β-tubulin was used as loading control. Source data are provided as Source Data files.

not depleted after irradiation (Fig. 7f), suggesting that the levels of the mitochondrial DNA polymerase rely also on factors other than DDR-dependent replicative senescence, in agreement with our previous observations (see Supplementary Fig. 2a, b).

Senescence can be induced in a DDR-independent manner through activation of the $p16^{Ink4}$ pathway[37]. Palbociclib, a compound that acts downstream of $p16^{Ink4}$ induces senescence by inhibiting cyclin-dependent kinases CDK4/6, and inducing permanent cell cycle arrest[38], see Fig. 7g. Early-passage (PN20) IMR-90 cells were treated with increasing doses of palbociclib (0.2–5 μM) for 7 days followed by 1 day of drug withdrawal. A dose-dependent increase of the fraction of SA-β-gal+ cells verified senescence (Fig. 7h, i), however, this did not result in increased γ−H2AX and 53BP1 foci number (Supplementary Fig. 9g–i), confirming that palbociclib-induced senescence is DDR-independent[39]. We observed decreased HTRA3 and increased CSB transcripts at all concentrations of the drug compared to untreated control (Fig. 7j, k), in clear contrast with replicative senescence (see Figs. 1c and 2a). Increased levels of CSB were confirmed by WB (Fig. 7m), which also showed unaffected HTRA3 levels despite reduced transcription. Palbociclib treatment did not increase the levels of $p16^{Ink4}$ transcript and protein, as expected since the drug acts downstream of p16 (Fig. 7l, m)[38]. Upon palbociclib treatment, p21 RNA and protein were upregulated (Fig. 7l, m), in agreement with a previous study[40]. Taken together, these data indicate that p16-dependent senescence does not correlate with depletion of CSB and overexpression of HTRA3, as it was the case for DDR-dependent/p21 replicative senescence.

**Delayed senescence with anti-ROS/anti-RNS treatment.** We showed previously that HTRA3 overexpression and POLG1 depletion in CS cells that lack CSB can be restored after 24 h treatment with MnTBAP, a superoxide dismutase mimetic and peroxynitrite (RNS) scavenger[22]. Incubating IMR-90 fibroblasts at various passages with MnTBAP for 24 h reduced SA-β-gal+ cells by 13.9 ± 6.0% at PN35 (Supplementary Fig. 10a). This reduction took place at constant cell number (Supplementary Fig. 10b), suggesting that it was not due to cell depletion. In agreement with our model, 24 h treatment with MnTBAP decreased the levels of HTRA3 compared to untreated controls, *de facto* blocking or containing HTRA3 overexpression in pre-senescent cells (PN27, PN31) and senescent cells (PN35), respectively (Supplementary Fig. 10c, d). Importantly, MnTBAP had essentially no effect on CSB levels (Supplementary

Fig. 10e, f), in agreement with the rescue of HTRA3/POLG1 levels in CS patient cells that are impaired in CSB.

Long-term treatment with a tenfold lower dose of MnTBAP resulted in increased cumulative population doubling of IMR-90 fibroblasts at PN31 and PN32 compared to untreated cells, suggesting a later entry into senescence in the presence of the drug (Supplementary Fig. 10g), which was confirmed by a decrease of SA-β-gal+ cells at PN29 and PN31 (≈35% and ≈9%, respectively; Supplementary Fig. 10h). Both effects were less pronounced than in cells treated with rapamycin, which extends replicative lifespan in vitro[41] (positive control). Thus, MnTBAP treatment delays replicative senescence of human fibroblasts without intervening on CSB levels.

## Discussion

The study of human ageing is expected to take advantage from model systems of monogenic progeroid syndromes such as Hutchinson-Gilford progeria syndrome (HGPS), Werner syndrome (WS), and Cockayne syndrome that mimic to some extent characteristics of normal ageing[42]. Indeed, in vitro and in vivo experiments performed on precocious ageing models showed that cellular and molecular hallmarks of ageing such as telomere dysfunction, epigenetic changes, metabolic defects or cellular senescence, recapitulate those of physiological ageing[43]. Moreover, despite in vitro reprogramming HGPS- or WS-derived iPSCs appeared to erase the pro-senescent phenotype, cells underwent again rapid and premature senescence after differentiation[18,44]. Based on our recent discovery that linked progeroid CS to improper degradation of functional mitochondrial proteins (POLG1 depletion via HTRA3/HTRA2 over-expression)[22], we proposed an alternative hypothesis, that CS-specific defects are implicated in the senescence of normal cells.

Serially passaged primary human fibroblasts display upregulation of the HTRA3 and HTRA2 proteases and downregulation of mitochondrial POLG1, suggesting a recapitulation of CS-specific defects[22] during replicative senescence. These defects include reduced mitochondrial respiration, in agreement with DDR-induced senescence[45], and depletion of mitochondrial complexes I, III, and IV. HTRA2 upregulation and POLG1 depletion were, however, not exclusive to cellular senescence but appeared to some extent also in long-term confluent cultured cells. This observation is in agreement with the finding that HTRA2 can induce cell cycle inhibition following serum withdrawal[46]. Conversely, HTRA3 upregulation appeared contemporary to the induction of the major effector of replicative

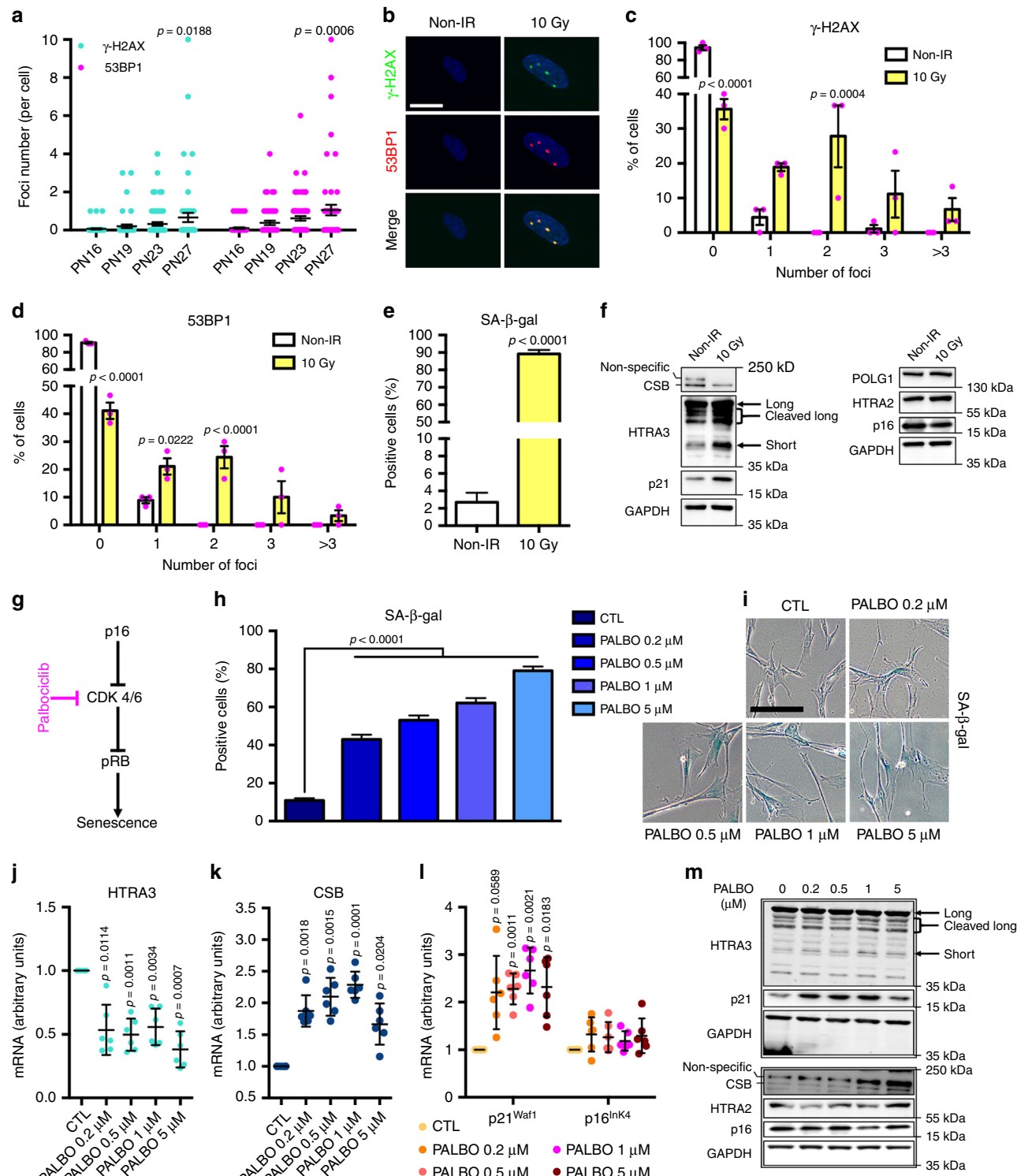

senescence p21, with comparable kinetics, thereby establishing this protease as a bonafide marker of cellular senescence.

HTRA3 is defined as a prevalent mitochondrial protein that is translocated from mitochondria upon cytotoxic stress, based on subcellular fractionation and colocalizaton experiments[26]. Our analysis with super-resolution SIM microscopy rather indicates that only a fraction of HTRA3 is mitochondrial. This condition is maintained in senescent cells, where accumulation of HTRA3 takes place in a mitochondrial-impoverished cytoplasm. HTRA3 accumulation during senescence concerned mainly the cleaved isoform of this protein, which is the more active isoform[27], suggesting an

increase of its auto-proteolytic activity during senescence. Secretion of HTRA3 was reported in the context of remodeling of the extracellular matrix or acting as a proapoptotic agent[26,47]. These observations raise the possibility that this protease can function as part of the SASP, and participate in tissue remodeling[11] or paracrine induction of senescence in neighboring cells[48].

Knockdown experiments demonstrated that contemporary accumulation of HTRA3 and the senescence inducer p21 during replicative senescence depend on depletion of CSB. We show that HTRA3 and p21 levels are inversely correlated with CSB, therefore, CSB depletion in normal cells is the earliest trigger of

**Fig. 7** CSB-dependent senescence is specific to the DDR/p21 pathway. **a** Enumeration of endogenous $\gamma-$H2AX and 53BP1 foci/cell at PNs preceding or during CSB depletion. $n = 50$–70 cells from three independent experiments; mean ± SEM; one-way ANOVA ($\gamma-$H2AX: $F = 3.209$, DFn = 3, DFd = 220, $p = 0.0239$; 53BP1: $F = 5.603$, DFn = 3, DFd = 220, $p = 0.001$) with post-hoc Tukey's test vs. the respective PN16. **b** Representative confocal acquisitions of irradiated (10 Gy) and non-irradiated (non-IR) IMR-90 immunostained for $\gamma-$H2AX (green) and 53BP1 (red), counterstained with Hoechst (blue, nuclei), after maximum intensity projection with the Imaris software; scale bar = 20 μM. Percentage of cells with 0, 1, 2, 3, or >3 (**c**) $\gamma-$H2AX or **d** 53BP1 foci per nucleus. $n = 90$ cells from three independent experiments. Mean ± SEM; two-way ANOVA ($\gamma-$H2AX: $F = 35.77$, DFn = 4, DFd = 20, $p < 0.001$; 53BP1: $F = 57.63$, DFn = 4, DFd = 20, $p < 0.0001$) with post-hoc Sidak's test vs. the respective non-IR. **e** Quantification of SA-β-gal$^+$ cells of irradiated and non-IR IMR-90 at PN17. $n = 180$–220 cells/condition from three independent experiments, mean ± SEM; unpaired Student's $t$-test (two-tailed) ($t = 36.28$, DF = 408), $p$-value vs. non-IR (**f**) Immunoblots of CSB, HTRA3, p21, and POLG1, HTRA2, p16 (GAPDH F-C staining, loading control, under each blot) in irradiated and non-IR fibroblasts. **g** Scheme indicating the palbociclib target in the p16 pathway to senescence. **h** Quantification and **i** representative images of SA-β-gal$^+$ staining of palbociclib-treated and untreated IMR-90 cells, scale bar = 200 μM. Early-passage fibroblasts (PN20) in normal medium (CTL) or in the presence of 0.2, 0.5, 1, or 5 μM palbociclib for 7 days followed by 1 day of drug withdrawal; $n = 800$ cells (CTL) and $n = 320$–380 cells (palbociclib-treated samples) from three independent experiments, mean ± SEM; one-way ANOVA ($F = 195.1$, DFn = 4, DFd = 2236, $p < 0.0001$) with post-hoc Tukey's test. RT-qPCR for **j** HTRA3, **k** CSB, and **l** p21$^{Waf1}$ and p16$^{Ink4}$. $n = 3$ independent experiments, mean ± SD; RM one-way ANOVA (HTRA3: $F = 53.07$, DF = 4, $p < 0.0001$; CSB: $F = 29.47$, DF = 4, $p < 0.0001$; p21$^{Waf1}$: $F = 13.36$, DF = 4, $p = 0.0015$; p16$^{Ink4}$: $F = 1.587$, DF = 4, $p = 0.2531$) with post-hoc Tukey's test vs. CTL. **m** WB of HTRA3, HTRA2, p21, p16, and CSB. Samples on the same blot are framed; each frame displays the respective GAPDH (F-C staining) used as a loading control. Source data are provided as Source Data files.

replicative senescence defined to date, acting upstream of p21$^{Waf1}$ induction. Importantly, despite loss of CSB downregulation upon passage in culture, the initial silencing was sufficient to reduce the replicative lifespan of cells by several weeks.

p21 induction in senescence is normally triggered by p53, a master regulator of the response to stress and DNA damage[11]. However, the role of p53 in replicative senescence has not been fully clarified, since for instance p53 does not promote this process in normal human keratinocytes[33]. By silencing or inhibiting p53, alone or together with CSB, we showed first that CSB-mediated senescence does not depend on p53. Moreover, p53 silencing alone had virtually no impact on replicative senescence, at least at early PNs. One may expect increased proliferation rate upon p53 silencing, however, it is possible that during early passages the extent of DNA damage is lower than required to trigger the p53 response and thereby cell cycle arrest.

These data indicate that CSB, and not p53, plays a master role in replicative senescence of human fibroblasts. Previous studies in cells derived from CS patients highlighted the physical interaction of CSB with p53[49] where CSB participates in p53 poly-ubiquitination and degradation upon UV induction[50]. Consequently, CS patient cells have higher basal and UV-induced levels of p53 compared to wild type cells. Thus, despite increased p53 levels during replicative senescence that are compatible with protein stabilization (also depends on reduced CSB), this process is largely p53-independent. The strong inverse correlation between p21$^{Waf1}$ and CSB indicates that in our model system p21-dependent replicative senescence is triggered by CSB depletion rather than p53 induction, revealing a CSB depletion/p21$^{Waf1}$ axis in cellular senescence (Fig. 8). In agreement with this model, we showed direct interaction of CSB on the endogenous and an ectopic p21$^{Waf1}$ promoter and loss of this interaction when CSB is silenced or depleted following irradiation, suggesting that CSB protects the p21$^{Waf1}$ promoter from activation by other factors, which essentially do not include p53. Moreover, p53 is activated by DNA damage and DNA damaging signaling, which do not increase upon CSB silencing. These data point to CSB directly controlling senescence via p21 and HTRA3 upregulation rather than CSB depletion being a secondary effect of DDR-induced senescence.

CSB is a DNA repair protein but also acts as chromatin remodeler and transcription factor[24,51,52]. In addition to direct intervention at p21$^{Waf1}$ promoter, CSB may also promote p21 degradation through ubiquitylation due to its E3 ligase activity, as it has been hypothesized in CS cells[53]. In other contexts than replicative senescence, p21 can be upregulated in the presence of high levels of CSB, as it is the case upon palbociclib treatment, in agreement with a previous study where the process was reported to be p53-independent and rather relying on myc derepression[40]. However, in this case p21 upregulation did not precede senescence, and appeared in the same timeframe as SA-β-gal staining, suggesting that it rather plays a role in the maintenance of senescence.

We demonstrate that CSB downregulation depends on histone H3 hypoacetylation of the CSB promoter, perhaps performed by various HATs, rather than combined hypermethylation and histone deacetylation of the CSB promoter as in UV-induced age-related nuclear cataract[31]. However, the DNA methylation status at other sites may affect CSB levels, since treatment with high doses of the methyltransferase inhibitor 5-aza-dC resulted in increased levels of CSB. Ectopic overexpression of CSB in a fraction of cells resulted in delayed signs of senescence, but in the long-term CSB overexpression may reduce or block p21 expression, and p21 is essential for the survival of senescent cells[54]. Compatibly, CSB overexpression was observed in cancer cells[55] that do not undergo senescence.

Replicative senescence involves permanent activation of the DDR/p53/p21 pathway[10]. We showed that CSB depletion and CSB promoter hypoacetylation coincide with increased DNA damage (DSBs) during replicative senescence. DNA damage signal though telomere attrition is unlikely to play a role, since replicative senescence of IMR-90 fibroblasts in 20% oxygen has been shown to be telomere-independent[56]. Despite CSB is also implicated in DSB repair through chromatin remodeling[57], it is unlikely that this activity affects the early phases of replicative senescence since CSB depletion is simultaneous to increase of DNA damage. However, severe CSB depletion at later PNs may impair DSB repair, contributing to exacerbate the senescent phenotype. Our findings indicate that CSB depletion and subsequent HTRA3 overexpression in replicative senescence are specific to the DDR/p53/p21 pathway (they are also activated by radiation (DDR)-induced senescence), and not to DDR-independent/p16 senescence pathway.

Transcription of CSA, the other major protein mutated in CS, and which is not associated with the most severe form of the disease[58], is not altered with replicative senescence, but the protein is depleted, although at a lower extent and several passages later than CSB, suggesting a different effect on senescence than CSB. Interestingly, we showed CSA depletion upon efficient silencing of CSB, suggesting that CSA levels depend, at least in part, on CSB. CSA and CSB are both involved in TC-NER and BER[59,60]. The apparently different underlying mechanisms and

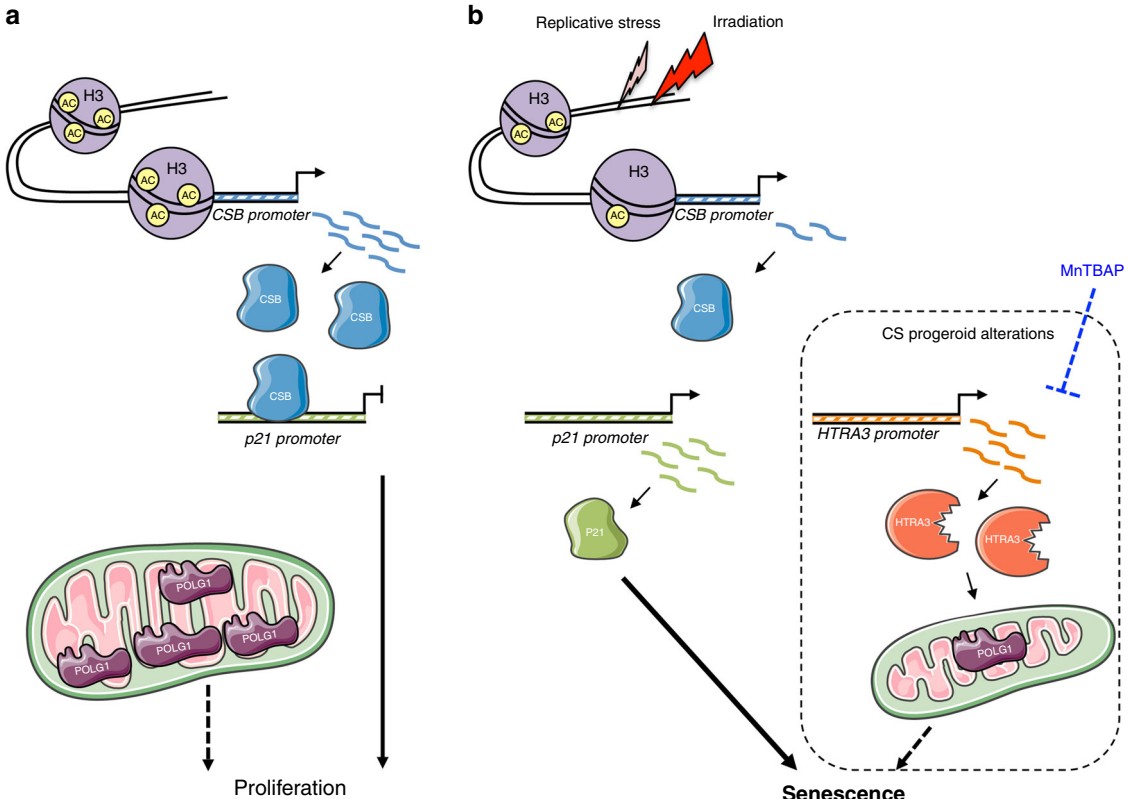

**Fig. 8** Scheme: role of CSB depletion in replicative senescence. Illustration of epigenetic regulation of CSB expression and its effect on downstream targets (**a**) regular CSB expression (**b**) low CSB levels leading to replicative senescence. Novel senescence effectors that are shared with the Cockayne syndrome paradigm (a disease due to CSB impairment in its most severe form) are shown in a framed area. Transcripts are indicated with a wavy line, active promoter with an arrow; Ac, acetyl group.

perhaps impact of CSB and CSA in replicative senescence support the notion that other causes than just impaired DNA damage repair are responsible for the severe defects of this disease[22,53].

Finally, a 24 h treatment with MnTBAP limited HTRA3 overexpression in replicative senescence, but had no effect on the senescence trigger CSB, and in the long-term increased replicative lifespan of fibroblasts. This result indicates that MnTBAP acts downstream of, and partially compensates for, defects triggered by CSB depletion, in agreement with rescue of the HTRA3/POLG1 defects in CS cells that are CSB impaired[22]. A previous study demonstrated that MnTBAP partially inhibits radiation-induced senescence in endothelial cells by rescuing complex II dysfunction[61]. These results suggest that MnTBAP limits the paracrine propagation of senescence by restricting the HTRA3/POLG1/mitochondrial dysfunction, with implications for the treatment of CS, as well as ageing prevention therapies.

Our data reveal a role of CSB in protecting human primary fibroblasts from DDR-dependent replicative senescence, via downregulation of the senescence trigger p21, and link a progeroid factor to physiological ageing. This pathway points to CSB as a relevant functional anti-ageing regulator. In this context, cellular senescence could also participate in the precocious ageing phenotype observed in CS, as it is suggested by premature senescence of patient-derived *CSA*-mutated keratinocytes[21]. It should be investigated in the future at which extent this defect, if confirmed in all CS patients, is correlated with and participates to the severity of the disease, whether CSA is implicated in the same pathway as CSB, and also whether it is operating in UVSS patients that lack the progeroid phenotype and whose cells do not display the HTRA3/POLG1/mitochondrial defect[22]. Finally, these data provide a DNA repair-independent mechanistic explanation for the reported association between high CSB levels and tumor proliferation[55], and CSB polymorphism association with longevity[62], supporting a key regulatory role of CSB in the ageing/cancer axis.

## Methods

**Cell culture, population doubling, and treatments**. Normal female human fœtal lung IMR-90 fibroblasts were obtained from ATCC (CCL-186™). IMR-90 cells (initially thawed at PN14) were cultured in minimum essential medium (MEM, Gibco) supplemented with 2 mM L-glutamin (GlutMAX), 10% fetal bovine serum (FBS, Gibco; LOT#42Q7051K), 1% Penicillin–streptomycin (Gibco), 1% non-essential amino acids (Gibco) and 1% sodium pyruvate (Gibco) in 20% $O_2$/5% $CO_2$ at 37 ℃. When indicated, cells have been thawed at a later passage than PN14. When indicated, cells were treated with increasing doses (0.2, 0.5, 1 and 5 μM) of pablociclib (Selleckchem, S1579), a cdk4/6 inhibitor, for 7 days followed by 1 day of drug withdrawal, or with 100 μM of MnTBAP (Merck, 475870), a reactive oxygen/nitrogen species (ROS/RNS) scavenger for 24 h. Early-passage IMR-90 cells (PN16) were harvested after 24 h (or 48 h when indicated) treatment with different histone acetyltransferase (HAT) inhibitors: 1–8 μM anacardic acid (AA) (Sigma-Aldrich, A7236, diluted in DMSO), 50 μM MG149 (Sellckchem S7476, diluted in DMSO), 50 μM CPTH2 (Focus Biomolecules, 10-4528, diluted in DMSO), 20 μM C646 (Sigma-Aldrich, SML0002, diluted in DMSO) or 10 μM curcumin (Sigma-Aldrich, C1386, diluted in ethanol), or with 5 μM MS275 (Sellckchem S1053, diluted in DMSO), a histone deacetylase (HDAC) inhibitor, or with increasing doses (2 and 10 μM) of 5-aza-2'-deoxycytidine (5-Aza-dC, Enzo Life Sciences, AXL-480-096-M005), a DNA methyltransferase inhibitor. Experiments with AA have been performed for 24 h and then cells were harvested, or cultivated for further 24 h upon drug withdrawal before harvesting. Control experiments were performed in the presence of an equivalent amount of the respective drug dilution buffer. When indicated, early-passage IMR-90 fibroblasts (PN18) were either expanded or left growing until confluence and either harvested (Conf) or kept for one or two more days in culture with medium changed daily (Conf + 1D and Conf + 2D, respectively) to ensure they stopped proliferating, before harvesting.

For replicative senescence, IMR-90 (PN15) were grown to 80% confluence, counted, and serially subcultured (dilution 1:3) until cessation of growth. At each PN, the population doubling (PD) was calculated with the following formula: 3.32 (log (total viable cells at harvest/total viable cells at seed)); cumulative PDs were then plotted against time. In further experiments, IMR-90 fibroblasts were harvested at PN16, PN19, PN23, PN27, PN31, and PN35. Successive passages in culture were also applied to IMR-90 (PN25), in presence or absence of 10 μM of MnTBAP or 10 nM rapamycin (diluted in DMSO, Enzo Life Sciences, BML-A275) in the cultured medium, and cells were harvested at PN27, PN29, and PN31 for further analyses.

For radiation-induced senescence, IMR-90 (PN16) were grown to confluence and irradiated at 10 Gy with the Xstrahl RS320 irradiator (X-ray). After irradiation, the medium was refreshed and cells were incubated overnight before 1:3 dilution and cultured for 10 additional days before experiments, according to ref. [35]. When indicated, cells were harvested at 7 and 15 days post-irradiation.

**SA-β-gal staining.** Cells were fixed with 2% paraformaldehyde (PFA, Alfa Aesar, 43368) and 0.2% glutaraldehyde (Sigma-Aldrich, G6257) in phosphate-buffered saline (PBS, Gibco) for 5 min at room temperature (RT) and washed twice with PBS before overnight staining at 37 °C with a 1 mg.ml⁻¹ X-Gal solution (Thermo-Fisher Scientific, 15520018) containing 5 mM $K_4[Fe(CN)_6]$, 5 mM $K_3[Fe(CN)_6]$, 150 mM NaCl, 2 mM $MgCl_2$, 40 mM citric acid/sodium phosphate, at pH 6.0. Cells were then rinsed in PBS and stored at 4 °C until imaging by light microscopy. A 500–1500 cells were assessed for each sample.

**Lentiviral particles production and shRNA-mediated knockdown.** Lentiviral particles were produced in HEK-293T cells (Institut Pasteur) after transfection of the following constructs: pLKO.1_puro encoding shRNA directed against human CSB (shCSB#1: Sigma-Aldrich, TRCN0000436471 and shCSB#2: Sigma-Aldrich, TRCN0000431424) or p53 (shp53: Sigma-Aldrich, TRCN0000003753), with the help of additional psPAX2 (packaging) (Addgene, 12260) and pMD2.G (envelope) (Addgene, 12259) plasmids, using calcium phosphate transfection method. The pLKO.1_puro Non-Mammalian shRNA (shSCR: Sigma-Aldrich, SHC002) was used as negative control. The culture medium was harvested and concentrated 48 h post-transfection.

Transduction of IMR-90 (PN18 or PN19) primary fibroblasts with shSCR, shCSB#1 or shCSB#2 was performed by incubating cells overnight with the viral particles in presence of polybrene (8 μg ml⁻¹) (Sigma-Aldrich, H9268). After 48 h, cells were resuspended in fresh puromycin (Toku-E, P001)-containing medium (2 μg.ml⁻¹) and selected for another 48 h. Surviving cells were replated at the same density (8000 cells.cm⁻²) and subjected to replicative senescence. Successive passages in culture were also applied to transduced IMR-90, in presence or absence of 10 μM of pifithrin-α (PFT-α) (diluted in DMSO, Sigma-Aldrich, P4359) in the culture medium. For knockdown of both p53 and CSB, IMR-90 (PN19) were first transduced with shp53, selected with puromycin as indicated above, and transduced a second time with shCSB (#2).

**Transfection of promoter reporter and expression vectors.** The *CDKN1A* (encoding *p21waf1*) promoter-eGFP reporter plasmid (pEZX-PF02) (HPRM51895-PF02, GeneCopoeia) was transfected in early-passage IMR-90 cells (PN15) using Lipofectamine 3000 (ThermoFisher) according to the manufacturer instructions. Forty-eight hours post-transfection, cells were cultured with 2 μg.ml⁻¹ puromycin for >2 weeks to select stable integrations. The resultant stable cell population expressing GFP under the control of the *CDKN1A* promoter was then either not transduced or transduced with shSCR or shCSB#2, as described in the previous paragraph.

Ectopic expression of CSB was obtained upon transfection with Lipofectamine 3000 of early-passage IMR-90 (PN15) with the *CSB* (Myc-DDK-tagged) expression vector (pCMV6-Entry) (RC219020, OriGene), called here "pCSB". Transfection with pcDNA3.1 (ThermoFisher Scientific, V79020), which codes for neomycin resistance under control of a CMV promoter or "Empty vector" was used as control. The CSB coding sequence was verified by sequencing pCSB before transfection. pCSB was amplified in Stbl2™ *E.coli* (ThermoScientific) at 30 °C, since amplification of pCSB in classic DH5 α *E.coli* resulted in multiple mutations (as indicated by the supplier). Transfected cells were kept in culture in the absence of selection for 8 days, including a passage in culture, before analysis.

**RNA isolation and real-time quantitative PCR (RT-qPCR) assay.** Total RNA was isolated from cells using the RNAeasy® Micro kit (Qiagen) and reverse-transcribed in cDNA with SuperScript IV Reverse Transcriptase (Thermo-Fisher Scientific). Possible RNA residues were degraded by incubation for 20 min at 37 °C with RNaseH (Takara Bio, 2150B). RT-qPCR was performed using PowerUp™ SYBR™ Green Master Mix (Thermo-Fisher Scientific) and the rate of dye incorporation was monitored using the StepOne Plus RealTime PCR system (Applied Biosystems). Three independent replicates were used for each condition. Data were analyzed by StepOne Plus RT PCR software v2.1 (Thermo Fisher Scientific). *TBP* (human) transcript levels were used for normalization of each target (=ΔCT). Real-time PCR $C_T$ values were analyzed using the $2^{-\Delta\Delta Ct}$ method to calculate the fold expression[63]. For experiments with palbociclib the fold expression was calculated

individually for each replicate. qPCR primers used are listed in Supplementary Table 1, which includes the corresponding references.

**Immunostaining.** Cells plated on glass slides were fixed with 2% (wt/vol) PFA and permeabilized with 0.5% Triton X-100 (Sigma-Aldrich, T9284) for 15 min. The glass slides were incubated overnight in blocking solution (5% BSA (wt/vol) (Gerbu Biotechnik, 1503) in PBS) at 4 °C then 1 h at room temperature with the primary antibody (α-HTRA3 (1:200, Sigma-Aldrich, HPA021187), α-HTRA2 (1:200, Sigma-Aldrich, HPA027366), α-POLG1 (1:200, Santa Cruz Biotechnology, sc-48815), α-CSB (1:200, Euromedex, CSB-3H8), α-Histone H2A.XpSer139 (1:2500, Millipore, 05-636), α-53BP1 (1:2000, Novus, NB100-304), α-p21 (1:500, BD Biosciences, 556430), α-ATP synthase β (1:200, Life technologies, A21351), α-GFP (1:1000, Abcam, ab13970), in a humidified chamber. After washing three times in PBS, samples were incubated with 10 μg.ml⁻¹ Hoechst 33342 (Sigma-Aldrich, 14533) and the appropriate Alexa 488- or 555-conjugated secondary antibody (1:1000, Thermo Fisher Scientific) for 1 h at room temperature.

For the immunofluorescence assessment of global DNA methylation levels, cells were treated as in ref. [64]. Briefly, cells on glass slides were fixed with 0.25% (wt/vol) PFA (10 min, 37 °C) and 88% methanol (30 min, −20 °C). Cells were incubated in 1 N hydrochloric acid (30 min, 30 °C) to achieve antigen retrieval, and neutralized with 0.1 M sodium borate (pH 8.5). Glass slides were then incubated for 20 min in blocking solution (2% goat serum (wt/vol) Abcam, ab7481, in PBS) at 37 °C then overnight at 4 °C with the primary antibody α-5-methlycytosine (1:2000, Merck, MABE146, clone 33D3). Finally, samples were incubated with 10 μg.ml⁻¹ Hoechst 33342 and the appropriate Alexa 555-conjugated secondary antibody for 1 h at room temperature.

**Confocal acquisition and quantification.** Confocal acquisitions were performed using a spinning-disk Perkin-Elmer Ultraview RS Nipkow Disk, an inverted laser-scanning confocal microscope Zeiss Axiovert 200 M with a 40×/1.4 NA objective and a Hamamatsu ORCA II ER camera (UTechS PBI of Institut Pasteur, Paris). Optical slices were taken each 200-nm interval along the z-axis covering the whole depth of the cell, at resolution of 1.024/1.024 pixels[22]. Three-dimensional (3D) reconstruction was achieved using the Imaris software (Bitplane). Fluorescence quantification corresponding to the sum of the gray values of all the pixels in each cell divided by the number of pixels per cell and cell surface measurement were done with the ImageJ 2.0 v software (NIH) (this analysis normalizes the number of pixels with the cell size, and is relevant for comparing values of senescent cells that increase in size). For each condition, 30–50 cells were analyzed from three independent experiments.

**Structured illumination microscopy (SIM).** IMR-90 fibroblasts at PN16 and PN35 were cultured on precision glass coverslips thickness No. 1.5 H (Marienfeld Superior). Immunofluorescence was performed as indicated in the previous section, using Fluoromont-G (Thermo fisher Scientific) as mounting medium. SIM was performed on a Zeiss LSM 780 Elyra PS1 microscope (Carl Zeiss, Germany) using 63×/1.4 oil Plan Apo objective with a 1.518 refractive index oil (Carl Zeiss) and an EMCCD Andor Ixon 887 1 K camera for the detection. Fifteen images per plane per channel (five phases, three angles) were acquired with a Z-distance of 0.09 μm to reconstruct 3D-SIM images. Acquisition parameters were adapted from one image to one other to optimize the signal to noise ratio. SIM images were processed with ZEN software and then aligned with ZEN using 100-nm TetraSpeck microspheres (ThermoFisher Scientific) embedded in the same conditions as the sample. The SIMcheck plugin in imageJ was used to analyze the acquisition and the processing in order to optimize for resolution, signal to noise ratio, and reconstruction pattern[65].

**Western blotting.** Cells were lysed with lysis buffer (50 mM Tris-HCl pH 7.5, 150 mM NaCl, 1% Triton X-100, 0.1% SDS, 1 mM EDTA, and protease/phosphatase inhibitor mixture (Roche)). Lysed cells were not centrifuged, and the whole extract was subjected to sonication (Bioruptor, Diagenode). The protein content was determined with the Bradford assay, and 10 μg of protein were boiled in the presence of NuPAGE LDS sample buffer (Invitrogen NP0007) and NuPAGE Sample Reducing Agent (Invitrogen NU0004) at 95 °C for 5 min, run for SDS-PAGE (4-12% Bis-Tris Gel, Thermo-Fisher Scientific) and transferred on a nitrocellulose transfer membrane (Bio-Rad). The membrane was then blocked with 5% BSA in phosphate buffer saline (PBS)-0.1% Tween20 (Sigma-Aldrich, P1379) for 1 h at RT and probed with specific primary antibodies (α-HTRA3 (1:1000, Sigma-Aldrich, HPA021187), α-HTRA2 (1:500, Sigma-Aldrich, HPA027366), α-POLG1 (1:1000, Santa Cruz Biotechnology, sc-48815), α-CSB (1:2000, Bethyl, A301-345A), α-CSA (1:1000, Abcam, ab137033), α-p21 (1:500, BD Biosciences, 556430), α-CDKN2A/p16INK4a (1:1000, Abcam, ab54210), α-p53 (1:1500, Santa Cruz Biotechnology, sc-126), α-PCNA (1:200, Santa Cruz Biotechnology, sc-56), α-Cyclin A2 (1:2000, Abcam, ab181591), α-phospho-ATM (1:500, Rockland Immunochemicals, 200-301-400), α-Histone H2A.XpSer139 (1:500, Millipore, 05-636), α-GFP (1:1000, Rockland Immunochemicals, 600-401-215), α-GAPDH (1:4000, Santa Cruz Biotechnology, sc-25778), hFAB Rhodamine α-GAPDH (1:5000, Bio-Rad, 12004168), hFAB Rhodamine α-β-Tubulin (1:5000, Bio-Rad, 12004166), α-Histone H3 (1:1000, Abcam, ab1791), α-Histone H3ac (1:5000,

Millipore, 06-599) overnight at 4 °C. After five washes in PBS containing 0.1% Tween20, the membrane was incubated with HRP-conjugated (1:10000, Thermo Fisher Scientific) or fluorophore-conjugated (F-C) secondary antibodies (1:5000, Bio-Rad and 1:25000, Biotium) for 1 h at RT and revealed by chemiluminescence or fluorescence respectively. When necessary, 0.1 M of glycine (pH 2) was used as stripping buffer for 20 min, followed by reblotting with HPR-conjugated secondary antibodies. Detection was performed using Chemidoc™ MP imaging system (Bio-Rad). Experiments were done in duplicate, and a representative Western blot was shown. Staining with ATX Ponceau S Red (Sigma-Aldrich, 09189) was used as a further marker of protein content. For Western blot analysis of the five OXPHOS complexes, cell lysates were not boiled and membranes were probed with the Total OXPHOS Human WB Antibody Cocktail (1:1000, Abcam, ab110411). Western blot bands quantification were done using the ImageJ software. Uncropped and unprocessed scan of all the blots are provided in the Source Data File.

**Quantitative chromatin immunoprecipitation (qChIP) assay**. For ChIP, $8 \times 10^6$ IMR-90 were cross-linked in 1% formaldehyde (Sigma-Aldrich, 252549) for 10 min, followed by addition of 125 mM glycine to stop the reaction (5 min). Cells were then washed in PBS, resuspended in lysis buffer (10 mM Tris-HCl (pH 8), 10 mM EDTA, 0.5 mM EGTA, 0.25% (v/v) Triton X-100, and protease and RNase inhibitors) for 5 min on ice. The pellets were then resuspended in 250 mM NaCl, 50 mM Tris-HCl (pH 8), 1 mM EDTA, 0.5 mM EGTA, for 30 min on ice to wash out non-cross-linked material. The resulting pellets were resuspended in 10 mM Tris-HCl (pH 8), 1 mM EDTA, 0.5 mM EGTA, 1% (w/v) SDS and chromatin was sheared by sonication (final average size of 200-500 base pairs, (bp)). The chromatin was diluted to obtain the following buffer composition: 0.1% (w/v) SDS, 1% (v/v) Triton, 0.1% (w/v) sodium deoxycholate, 10 mM Tris-HCl (pH 8), 150 mM NaCl, 1 mM EDTA, 0.5 mM EGTA, and protease and RNase inhibitors.

ChIPs were carried out by incubating 10 to 20 μg of chromatin with 1 μg of either antibody: (α-acetlyl-Histone H3 (Millipore, 06-599), α-Histone H3 (Abcam, ab1791), (N- and C- terminal) α-CSB (Bethyl, A301-345A and Abcam, ab96089, respectively), or α-rabbit IgG (Santa Cruz Biotechnology, sc-2027) as a control. After overnight incubation at 4 °C, 40 μl of protein G-Dynabeads™ (Thermo-Fisher Scientific, 10004D) were added on lysate for 2 h at 4 °C. After extensive washing, DNA was isolated from the beads by successive boiling for 10 min in the presence of 10% (w/v) Chelex 100 Resin (Bio-Rad, 1421253), incubating at 55 °C for 30 min in the presence of 100 μg.ml⁻¹ of Proteinase K (Eurobio Ingen, GEXPRK01-E), and boiling again for 10 min. After centrifugation, the resulting supernatant was used as a direct template for qPCR detection of regions of CSB (Supplementary Fig. 9c) or $p21^{Waf1}$ promoters (Fig. 4b) (primers listed in Supplementary Table 1). Immunoprecipitated chromatin with the indicated antibodies was calculated as a percentage of the input DNA after normalization with control ChIP performed with rabbit IgGs (Mock). Immunoprecipitated chromatin with acetyl-Histone H3 was finally normalized to the H3 histone. Chromatin templates were also used for PCR detection of regions of the promoter of $p21^{Waf1}$ (Fig. 4a) (primers listed in Supplementary Table 1), using the following PCR profile: preincubation for 5 min at 95 °C, 40 cycles of 30 s at 95 °C, 30 s at a primer-specific annealing temperature (60 °C) and 30 s at 72 °C, and one final 10 min incubation at 72 °C. The PCR products were separated by electrophoresis through 2.0% agarose gel and detected with the Gel Doc 2000 imaging system (Bio-Rad).

**Genomic DNA isolation and methylation assays**. Genomic DNA (gDNA) from IMR-90 was isolated using the QIamp® DNA mini kit (Qiagen) and 800 ng of DNA was treated with sodium bisulfite using the EZ DNA Methylation™ Kit (Zymo Research), according to supplier instructions, to convert non-methylated cytosine residues to uracile. Universal methylated human DNA standard (Enzo Life Sciences, ENZ-45005-0001) was use as a positive control.

The region of interest corresponding to a CpG island located in the promoter of the CSB gene (Supplementary Fig. 7) (from −603 to −396, transcription start site (TSS) as +1: chr10:50747072 (GRCh37 coordinates)) was PCR amplified using 20 ng of bisulfite-treated genomic DNA. The primers for CSB (listed in Supplementary Table 1) was designed by web-based MethPrimer 2.0 software[66]. The PCR products (208 bp) were gel extracted using the Wizard® SV Gel and PCR Clean-Up system (Promega) following the supplier instructions and sent for direct sequencing (Cochin Sequencing Platform, Paris, France). The level of methylation, expressed as a percentage of methyl-cytosine (%Methyl-C = (Methyl-C/(Methyl-C + C)) × 100), was analyzed at the single nucleotide resolution using the 4peaks software (Nucleobytes), in a total of eight CpG sites located at −564 (#1), −557 (#2), −545 (#3), −536 (#4), −528 (#5), −505 (#6), −494 (#7), −441 (#8) from the transcription starting site (TSS).

**ROS detection**. The levels of oxidative stress were assessed measuring reduced glutathione (GSH), a strong scavenger of ROS, and its ratio with oxidized glutathione (GSSG), using the GSH/GSSG-Glo™ Luminescent assay (Promega) following the manufacturer's instruction. The GSH/GSSG ratio was assessed in 20,000 cells plated on 96-well white-wall plate using the Tecan plate reader. For detection of mitochondrial superoxide ions, cells were incubated with 5 μM MitoSOX Red Mitochondrial Superoxide indicator (Thermo-Fisher Scientific, M36008) for 10 min at 37 °C.

**Total ATP steady-state assays**. Ten thousand cells plated on 96-well white-wall plate were treated with 10 μM oligomycin for 1 h (for glycolytic ATP) or untreated (for total ATP levels); cells were then tested with the CellTiter-Glo Luminescent assay (Promega) according to supplier instructions using a Tecan plate reader.

**Statistical analysis**. To determine statistical significance between groups, Student's $t$-test (unpaired, two-tailed; for two groups) and one-or two-way analysis of variance (ANOVA; for three or more groups) with post-hoc Tukey's or Sidak's test, were used assuming normal distribution of the data. For experiments with palbociclib the statistical significance of RT-qPCR was determined using RM (repeated measures) one-way ANOVA with post-hoc Tukey's test. Linear regression was modeled for selected pairs of transcripts using the R software. All statistical details of experiments, including the statistical test used, the number of independent experimental replications, the definition of center and dispersion, the relevant values of the test summary: for ANOVA the $F$ ratio, the corresponding degrees of freedom (DF) for the numerator (DFn) and the denominator (DFd), and the ANOVA $p$-value (interaction $p$-value for two-way ANOVA), and for the Student's $t$-test the $t$ ratio and DF, are reported in each corresponding figure legend. The post-hoc $p$-values are indicated in the figures panels: the exact $p$-value is indicated with the exception of $p < 0.0001$ as these values are no further specified by GraphPad Prism 6.0; post-hoc $p$-values > 0.05 are prevalently not indicated. All statistical analyses were performed using the GraphPad Prism 6.0 Software (GraphPad software).

**Reporting summary**. Further information on research design is available in the Nature Research Reporting Summary linked to this article.

## Data availability

All data generated or analyzed during this study are included in this published article (and its supplementary information files). All data is available from the corresponding author upon reasonable request.

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

## Acknowledgements

We thank Sebastien Mella for help with linear regression analyses and advice on statistical tests, Alain Sarasin, Han Li, and Shahragim Tajbakhsh for comments on the manuscript, Laurent Chatre for discussion, and Martin Feelisch for advice on ROS. UTechS PBI is part of the France–BioImaging infrastructure network (FBI) supported by the French National Research Agency (ANR-10-INSB-04; Investments for the Future),

and acknowledges support from ANR/FBI and the Région Ile-de-France (program "Domaine d'Intérêt Majeur-Malinf") for the use of the Zeiss LSM 780 Elyra PS1 microscope. This work was supported by grants from DARRI-Institut Pasteur (PasteurInnov 14–152) and PTR-Institut Pasteur (2017–111).

## Author contributions

C.C. designed and conducted most experiments, analyzed results, and wrote the manuscript. C.F.M. performed and analyzed experiment with palbociclib, irradiation and CSB overexpression, and contributed to the global analysis of results. B.M. helped C.C. in performing some experiments. A.S. performed super-resolution microscopy experiments. M.R. designed experiments, supervised the work, analyzed results, and wrote the manuscript.

## Competing interests

The authors declare no competing interests.
