## [Peer Review File · Nature Communications]

Reviewers' comments:

Reviewer #1 (Remarks to the Author):

The manuscript entitled 'CSB promotor downregulation via histone H3 hypoacetylation is an early determinant of replicative senescence' by Crochemore et al investigates the role of CSB and p21 in the process of senescence. This is an important aspect in our understanding of CS as a progeroid syndrome. There are several points to be addressed.

The authors demonstrate that CSB depletion induces cellular senescence in normal female human foetal lung IMR-90 fibroblasts. Figure 2 shows that CSB depletion is accompanied with cellular senescence. But CSB induced senescence could be just an associated secondary effect due to accumulated DNA damage and DNA damage signaling, not necessarily involving p21 upregulation as major transducer of senescence signaling. The depletion of CSB can lead to accumulation of DNA damage which is a potent driver of cellular senescence. To show that CSB depletion directly controls senescence via p21 upregulation and HTRA3 upregulation but not via DNA damage signaling the authors should analyze mutation load due to CSB depletion and other DNA damage signaling components.

For a better functional connection of CSB expression and senescence, expression of CSB protein should be upregulated during the process of replicative senescence to interfere with senescence (with transfection of CSB expression vectors or other systems). The overexpression of CSB should be performed at time points when naturally CSB depletion and replicative senescence occurs. If CSB depletion is causative for senescence then overexpression and restoration of CSB protein production should also delay cellular senescence.

The authors state that CSB and not CSA is the major driver of senescence but recently it was shown that expression of CSA in keratinocytes protected from senescence and restored important factors of cellular senescence (secretion of senescence associated secretory phenotype) (Cordisco et al 2019). The authors should discuss this.

The authors analyzed methylation status of approximately 300bp near the transcription start site (Figure 3 and supplementary Figure 4) although transcription can be influenced by methylation in more distant areas. As mentioned by the paper they cite (Wang et al 2016) the authors should perform experiments with methyltransferase inhibitors like 5 aza dc (Wang et al 2016) to see if modulation of methylation also affects CSB transcription as already reported by Wang et al (Figure 7).

In Figure 3 the authors show that Histone acetyltransferase (HAT) inhibitor anarcadic acid (AA) induces H3 hypoacetylation and decreases CSB expression. However AA is a broad HAT inhibitor not only specific to H3 acetylation at sites of CSB promoter (page 9 line 6) (Hemshekar et al 2011; Eliseeva et al 2007). The authors should perform additional experiments either with another HAT inhibitor or with HDAC inhibitors to investigate histone H3 acetylation and its influence on CSB transcription more deeply.

According to the data from this manuscript, loss of CSB protein should lead to premature senescence in cells. If these in vitro findings have clinical relevance then a CSB depletion induced cellular senescence in a patient should also give rise to severe premature aging symptoms in these patients. However the authors need to reconcile the fact that patients with CSB mutations leading to complete absence of CSB protein show only mild CS symptoms (Hashimoto et al 2008) or the UV sensitivity syndrome with mild premature aging symptoms (Horibate et al 2004).

Although the authors showed that CS interacts within the p21 promoter (Figure 2j) more experiments with reporter constructs (Yasutake et al 2013) including the p21 promoter and CSB depletion or CSB upregulation would be helpful to support the model in Figure 7.

Minor points

It should be written if the values shown in Figure 1 a are representative of three independent experiments. Otherwise standard deviation has to be shown if these are mean values.

In Figure 1 c it would be interesting to see mitochondrial localization of HTRA3 in fibroblasts of different passage numbers.

In supplementary Figure 2f it is unclear how 100% of cellular ATP content in PN17 cells is glycolysis-derived while another 100% of cellular ATP content in PN17 cells is derived via oxidative phosphorylation.

In supplementary Figure 2 g the decrease of expression of complex I and II is hard to see.

At page 6 in line 22 the authors mention that CSA protein is not decreased during senescence but recently it was published that CSA protects also from cellular senescence (Cordisco et al 2019).

The authors should comment on this.

At page 7 in line 17 the authors should correct the mistake 'precious' senescence

Horibata K1, Iwamoto Y, Kuraoka I, Jaspers NG, Kurimasa A, Oshimura M, Ichihashi M, Tanaka K. Complete absence of Cockayne syndrome group B gene product gives rise to UV-sensitive syndrome but not Cockayne syndrome.

Proc Natl Acad Sci U S A. 2004 Oct 26;101(43):15410-5.

Wang Y, Li F, Zhang G, Kang L, Guan H.

Ultraviolet-B induces ERCC6 repression in lens epithelium cells of age-related nuclear cataract through coordinated DNA hypermethylation and histone deacetylation.

Clin Epigenetics. 2016 May 26;8:62.

Tetsuo Yasutake,1 Hiroo Wada,1 Manabu Higaki,1 Masuo Nakamura,1 Kojiro Honda,1 Masato Watanabe,1 Haruyuki Ishii,1 Shigeru Kamiya,1 Hajime Takizawa,a,1 and Hajime Goto1 Anacardic acid, a histone acetyltransferase inhibitor, modulates LPS-induced IL-8 expression in a human alveolar epithelial cell line A549

Version 1. F1000Res. 2013; 2: 78.

Published online 2013 Mar 6. doi: 10.12688/f1000research.2-78.v1

PMCID: PMC3931454

PMID: 24627774

Hemshekhkar M, Sebastin Santhosh M, Kemparaju K, et al. :

Emerging Roles of Anacardic Acid and Its Derivatives: A Pharmacological Overview.

Basic Clin Pharmacol Toxicol. 2011.

Cordisco S, Tinaburri L, Teson M, Orioli D, Cardin R, Degan P, Stefanini M, Zambruno G, Guerra L, Dellambra E.

Cockayne Syndrome Type A Protein Protects Primary Human Keratinocytes from Senescence.

J Invest Dermatol. 2019 Jan;139(1):38-50.

Hashimoto S, Suga T, Kudo E, Ihn H, Uchino M, Tateishi S.

Adult-onset neurological degeneration in a patient with Cockayne syndrome and a null mutation in the CSB gene.

J Invest Dermatol. 2008 Jun;128(6):1597-9.

Reviewer #2 (Remarks to the Author):

Chrochemore et al report that depletion of CSB due to H3 hypoacetylation causes upregulation of p21 and senescence. They also suggest that CSB depletion can promote mitochondrial impairments via overexpression of the HTRA3 proteases. The manuscript is technically ok, but many important controls are missing. Organization of figures and panels is sometime confusing, and there are several overinterpretation of data. While the role of CSB in promoting senescence is quite well demonstrated, the impairment of mitochondrial functions is definitely not evident.

Major points:

Fig 1C. What do the authors mean with 'independent of cell size'? Is this normalized somehow? If so, quantification of cell volume should be provided

Organization of panels in figure 1 is highly confusing

Quantification of global and mitochondrial ROS suggests that the difference between pre- and senescent cells is minimal at best. First, images of fluorescence intensity should be provided.

Second, more sensitive methods to measure ROS production/accumulation should be utilized

Figure 2e only suggests a slower population doubling time in shCSB cells. Also, no correlation between this figure and 2f and 2g, where senescence seems to happen much earlier. Authors should try to understand this discrepancy

It is not clear what is the phenotypical effect of using anacardic acid on proliferating cells

I would expect a more pronounced proliferation in cells without p53, which would be a good control for silencing. However, this doesn't seem to happen. Authors should discuss why

The effect of PFTa on p21 is not shown. Authors should provide evidences that the inhibitor is effective in these experiments

10 days after IR, p16 is normally upregulated in IMR90 cells (but not at earlier time points). It is surprising authors do not find elevated expression

Why p21 would go up independently of p53 and CSB upon palbociclib treatment? The conclusion of the figure is definitely not clear

Figure 6 is dispensable for the message of the paper, just confusing

We thank the referees for the assessment of our manuscript and for suggesting further experiments to consolidate our study. We have added 17 novel figure panels that represent new experiments (Fig. 2k-l; Fig. 4g and h; Supplementary Fig. 1c; Supplementary Fig. 4e and f; Supplementary Fig. 5b-h; Supplementary Fig. 6b and c), and two panels contain additional experiments (Fig. 1j and Fig. 2h). Moreover, we replaced Fig. 3h and Supplementary Fig. 3d with results from new experiments. Thus, totally we added 21 new experimental datasets.

We also added more information in Supplementary Fig. 3c, right panel, Supplementary Fig. 3e, lower panel, and Supplementary Table 2, and we changed the representation of data in Supplementary Fig. 3c, e, and f. The revised version contains one more supplementary figure (former Supplementary Fig. 1 has been split in the new version into Supplementary Fig. 1 and Supplementary Fig. 2 because of the additional panels), and as a consequence the number of the following Supplementary Figures has been shifted by +1.

[REDACTED]

As requested by the Nature Communications Editorial Policy Checklist, we indicated the metrics of the statistic tests (F, t, degrees of freedom, exact p-value) in the Figure legends and Material and Methods and, when applicable, modified the panel of the figures accordingly.

Reviewer #1 (Remarks to the Author):

The manuscript entitled 'CSB promotor downregulation via histone H3 hypoacetylation is an early determinant of replicative senescence' by Crochemore et al investigates the role of CSB and p21 in the process of senescence. This is an important aspect in our understanding of CS as a progeroid syndrome. There are several points to be addressed.

Response. We thank the Referee for the comment on the relevance of our study, and address below the points raised.

The authors demonstrate that CSB depletion induces cellular senescence in normal female human foetal lung IMR-90 fibroblasts. Figure 2 shows that CSB depletion is accompanied with cellular senescence. But CSB induced senescence could be just an associated secondary effect due to accumulated DNA damage and DNA damage signaling, not necessarily involving p21 upregulation as major transducer of senescence signaling. The depletion of CSB can lead to accumulation of DNA damage which is a potent driver of cellular senescence. To show that CSB depletion directly controls senescence via p21 upregulation and HTRA3 upregulation but not via DNA damage signaling the authors should analyze mutation load due to CSB depletion and other DNA damage signaling components.

Response. Figure 2 indeed showed a direct link between CSB depletion and senescence. Please note that senescence was accelerated despite transitory silencing of CSB, and also that silencing was extensive but not complete at these early time points, further underscoring the strong pro-senescence effect of CSB depletion. This Referee, however, wonders whether this effect could be rather a consequence of accumulated DNA damage and DNA damage signaling, not involving p21 upregulation. We are not aware of DNA damage-induced senescence that is independent of p21 upregulation, since DNA damage induces p53 that in turn induces p21 that blocks the cell cycle. Moreover, Figure 2g, h, shows upregulation of p21 transcript and protein upon CSB silencing. We have also shown (Fig. 4) that p53 silencing or inhibition does not prevent induction of senescence, rather supporting p21 regulation by CSB de-repression than p53 activation.

We nevertheless tested this hypothesis assuming that p21 (and HTRA3) upregulation, as well as CSB depletion itself, are secondary effects or consequences of DNA-damage induced senescence. We thus checked whether CSB silencing was associated with increased endogenous DNA damage. Supplementary Fig. 4e (new panel) [please note that the number of supplementary figures has changed compared to the previous version of the manuscript] shows that γ -H2AX as well as 53BP1 foci number/cell assessed by immunofluorescence did not change upon CSB silencing (with either shCSB#1 or shCSB#2). Moreover, Supplementary Fig. 4f (new panel) confirms no increase of γ -H2AX in WB, as well as the upstream DNA damage response factor p-ATM (phosphorylated ATM), under CSB silencing. These experiments were performed with fixed cells and cell pellets at PN19 and PN20 from the very same silencing experiment in Fig. 2 and Supplementary Fig. 4 [previously Supplementary Fig. 3]. Given the lack of increase of endogenous DNA damage/DNA damage signalling under these conditions, assessing the mutation load was not justified, in our view. Indeed, mutations

would result from increased DNA damage (which is not the case in our experiments) that was incorrectly repaired, and assessment requires whole genome sequencing at multiple cell passages since specific genes/loci that undergo mutation upon senescence have not been identified, overall an unnecessary effort given the abovementioned lack of DNA damage accumulation.

We conclude that CSB silencing is not associated with increased DNA damage, and rather results in p21 (and HTRA3) upregulation, as originally claimed. Therefore, CSB silencing is not a secondary effect, but rather a trigger, of DNA damage-induced senescence. Results are shown at page 8 and discussed at page 15.

For a better functional connection of CSB expression and senescence, expression of CSB protein should be upregulated during the process of replicative senescence to interfere with senescence (with transfection of CSB expression vectors or other systems). The overexpression of CSB should be performed at time points when naturally CSB depletion and replicative senescence occurs. If CSB depletion is causative for senescence then overexpression and restoration of CSB protein production should also delay cellular senescence.

Response. We fully appreciated the rationale of this request, which was, however, more challenging than expected to address. Although CSB overexpression has been observed in cancer cells (Caputo et al., 2013), and higher levels of CSB than in CSB-proficient control cells have been obtained upon reversal from silencing (in HeLa cells, (Chatre et al., 2015)) or expression in SV-40 immortalized CSB-deficient cells (e.g. (Selzer et al., 2002), CSB is possibly deleterious at high levels in normal primary cells (IMR-90). Indeed, it has been demonstrated that p21 is essential for viability of senescent cells (Yosef et al., 2017). Now, in the present work we show that CSB expression blocks p21 expression, therefore high CSB levels would block p21 also in cells where this protein must be maintained to ensure survival.

[REDACTED]

[REDACTED]

we transfected early passage IMR-90 (PN15) to maximise the transfection efficiency, with either pCSB or the empty plasmid, and cultured cells without selection for 8 days, including passing them once, before analysis.

We [REDACTED] observed that IMR-90 transfected with pCSB had higher levels of CSB, reduced p21 expression (Supplementary Fig. 4g), and also displayed a lower fraction of SA- β -gal⁺ cells than those transfected with the empty plasmid (Supplementary Fig. 4h), suggesting that higher expression of CSB delays senescence. These results and the rationale are presented at pages 8-9, and discussed at page 16.

[REDACTED]

The authors state that CSB and not CSA is the major driver of senescence but recently it was shown that expression of CSA in keratinocytes protected from senescence and restored important factors of cellular senescence (secretion of senescence associated secretory phenotype) (Cordisco et al 2019). The authors should discuss this.

Response. Before discussing this point, we performed WB of CSA at different passages of IMR-90 (using frozen cell pellets at different PN from experiment in Fig. 1). We observed a decline in CSA levels in pre-senescent (PN31) and senescent (PN35) cells (lower panel added in Fig. 1j), which was not detected at the RNA level (previous Fig. 1h); as indicated in the figure legend, reduction in CSA normalized to GAPDH were confirmed for PN31 and PN35 in other blots, not shown. Thus, also CSA is depleted in senescence, although considerably less extensively and at least five passage later than CSB, *i.e.* at the same time as the rise of the other senescence markers *p21^{Waf1}*, *IL-6*, *HTRA3*, *HTRA2* (Fig. 1c-f). Interestingly, whereas CSB declines at the level of transcript, the decline in CSA appears more at the protein level.

These data are therefore in agreement with (Cordisco et al., 2019) (cited) that focused on CSA-mutated Cockayne syndrome cells, as we mentioned in the original version of our manuscript. Moreover, we assessed CSA levels upon CSB silencing (using cell pellets frozen from experiment in Fig. 2) and observed at PN20 a decrease with the most efficient silencing (shCSB#2), suggesting that CSA levels depend, at least in part, on CSB (see added lower panel in Fig 2h). We have modified this point in results (page 7) and discussion (page 17). We thank the Referee for pointing out this important aspect that prompted us to do more in-depth analysis and better clarify the condition of CSA in replicative senescence.

The authors analyzed methylation status of approximately 300bp near the transcription start site (Figure 3 and supplementary Figure 4) although transcription can be influenced by methylation in more distant areas. As mentioned by the paper they cite (Wang et al 2016) the authors should perform experiments with methyltransferase inhibitors like 5 aza dc (Wang et al 2016) to see if modulation of methylation also affects CSB transcription as already reported by Wang et al (Figure 7).

Response. We agree that transcription can be influenced also by methylation at more distant sites than ≈ 300 bp. We originally tested this region of the CSB promoter because hypermethylation of a single site therein resulted in CSB downregulation in age-related cataract (Wang et al, 2016, cited), and constitutes to date the only known

change in methylation that alters CSB transcription. As requested, we treated early-passage IMR-90 fibroblasts (PN16) with the methyltransferase inhibitor 5-aza-dC. Treatment with 10 μ M 5-aza-dC for 24h resulted in increase of CSB transcripts (Supplementary Fig. 5c, new panel) and protein (Supplementary Fig. 5d, new panel), as it was also the case for age-related cataract, indicating that global DNA hypomethylation, possibly at distant sites, increases the expression of CSB. To note, this was not the case with 2 μ M 5-aza-dC (Supplementary Fig. 5c,d new), which also resulted in global reduction of DNA methylation, although not as much as with 10 μ M 5-aza-dC (Supplementary Fig. 5b, new panel). We presented and discussed these data at pages 9 and 16. To note that upon treatment for 24h with 5-aza-dC, despite drug withdrawal cells undergo rapid senescence and therefore the possible effect of increased CSB expression cannot be further tested in these conditions. This observation is in agreement with our findings on CSB overexpression, on the dedicated paragraph above.

In Figure 3 the authors show that Histone acetyltransferase (HAT) inhibitor anarcadic acid (AA) induces H3 hypoacetylation and decreases CSB expression. However AA is a broad HAT inhibitor not only specific to H3 acetylation at sites of CSB promoter (page 9 line 6) (Hemshekar et al 2011; Eliseeva et al 2007). The authors should perform additional experiments either with another HAT inhibitor or with HDAC inhibitors to investigate histone H3 acetylation and its influence on CSB transcription more deeply.

Response. We treated early-passage IMR-90 fibroblasts (PN16) with additional HAT inhibitors, namely MG149 (an anarcadic acid derivative with sensitivity toward Tip60 and MOF HATs, which are also involved in DNA repair), CPTH2 (specific of Gcn5p HAT), C646 (a potent selective inhibitor of p300 HAT), and curcumin (specific of p300/CBP), each of them inhibit one or more of the main families of HATs (Dekker et al., 2014). Each tested HAT inhibitor reduced the levels of CSB, with the most efficient deacetylase activity (reduced H3Ac levels) resulting in larger CSB depletion (Supplementary Fig. 5e, new). We also treated cells with the HDAC inhibitor (MS275, a specific inhibitor of class I HDAC, also used by (Wang et al., 2016), and observed no change in CSB levels, despite a large increase in H3Acetylation (Supplementary Fig. 5f). There was no increase in CSB levels also for shorter treatment with MS275 (2h, 8h, 16h, not shown).

Since histone acetylation/deacetylation observed above were assessed at the global genome level, we also analysed the levels of acetylated histone H3 on the CSB promoter by ChIP with either type of inhibitors. We observed decreased H3 acetylation with C646 and CPTH2 HAT inhibitors (MG142 and curcumin levels were not statistically significant), and increased H3 acetylation with MS275. These data indicate that CSB levels are affected by histone hypoacetylation, as we previously demonstrated, which is perhaps performed by multiple HATs, but are not affected by hyperacetylation. These results are presented at page 10 and discussed at page 16 of the revised manuscript.

According to the data from this manuscript, loss of CSB protein should lead to premature senescence in cells. If these in vitro findings have clinical relevance then a CSB depletion induced cellular senescence in a patient should also give rise to severe premature aging symptoms in these patients. However the authors need to reconcile the fact that patients with CSB mutations leading to complete absence of CSB protein show only mild CS symptoms (Hashimoto et al 2008) or the UV sensitivity syndrome with mild premature aging symptoms (Horibate et al 2004).

Response. We reasonably expect that CSB mutant cells from CS patients display earlier senescence compared to CSB-proficient control cells. Several markers of senescence indicate that this was the case for CSA mutated keratinocytes ((Cordisco et al., 2019), cited).

[REDACTED]

[REDACTED]

We do [REDACTED] propose [REDACTED] to [REDACTED] modify the existing sentence at pages 17-18 with the following one: “*In this context, cellular senescence could also participate in the precocious ageing phenotype observed in CS, as it suggested by premature senescence of patient-derived keratinocytes (Cordisco et al., [REDACTED]*

It should be investigated in the future at which extent this defect, if confirmed in all CS patients, is correlated with and participates to the severity of the disease, whether CSA is implicated in the same pathway as CSB, and also whether it is operating in UVSS patients that lack the progeroid phenotype and whose cells do not display the HTRA3/POLG1/mitochondrial defect (Chatre et al., 2015)”. Indeed, UVSS1 cells do not display HTRA3 upregulation (and thereby POLG1 depletion and mitochondrial dysfunction), and it is possible that the HTRA3 promoter, as well as the p21 promoter, are differently sensitive to the absence of CSB in UVSS cells than the are in CS cells. [REDACTED] this topic requires a dedicated study.

Although the authors showed that CS interacts within the p21 promotor (Figure 2j) more experiments with reporter constructs (Yasutake et al 2013) including the p21 promotor and CSB depletion or CSB upregulation would be helpful to support the model in Figure 7.

Response. We believe that having shown direct interaction of CSB with the p21 promoter through ChIP at at least three sites and with multiple approaches (Figure2 j) is a direct evidence of this interaction. However, we performed additional experiments, as requested. For this, we transfected early-passage (PN16) IMR-90 with the human *CDKN1A* (coding for p21) promoter-eGFP reporter plasmid and isolated cells with a stable integration after more than 2 weeks of selection with puromycin. Transduction of the resulting GFP⁺ cell population with shCSB#2 resulted in higher GFP immunofluorescence (Fig. 2l) and immunoblot (Fig. 2m) signals compared to shSCR (scramble), indicating increased p21 expression upon CSB silencing, in agreement with our model.

We additionally performed ChIP with the N-term and C-term anti-CSB antibodies on several regions of the p21 promoter 10 days after irradiation, *i.e.* in cells undergone senescence and that express low levels of CSB. Under these conditions (Supplementary Fig. S6c) we observed loss of the CSB interaction on the 21 promoter, compared to non-irradiated cells, again in agreement with our previous findings.

Minor points

It should be written if the values shown in Figure 1 a are representative of three independent experiments. Otherwise standard deviation has to be shown if these are mean values.

Response. Values in Figure 1a are indeed the mean of three independent experiments where the SD is very low (graphically within the size of the symbol). The same situation occurred in Supplementary Fig. 3a. Mean and SD for each passage of both panels are now reported in Supplementary Table 2.

In Figure 1 c it would be interesting to see mitochondrial localization of HTRA3 in fibroblasts of different passage numbers.

Response. Based on immunoblotting of subcellular fractionation and co-localization experiments with classic microscopy, HTRA3 has been defined as a (prevalent) mitochondrial protease that is translocated from mitochondria upon cytotoxic stress (Beleford et al., 2010). Given the extensive cytoplasmic signal of HTRA3 in Fig. 1c we doubted that colocalization with mitochondrial markers would be resolvable with confocal spinning disk microscopy, and this was indeed the case (not shown). We thus performed super-resolution SIM microscopy of early passage (PN16) and senescent (PN35) IMR-90 fibroblasts and observed that in both cases HTRA3 is not prevalently mitochondrial but rather extra-mitochondrial, and several mitochondria appear lacking HTRA3 (new panel Supplementary 1c). We presented this data in results (page 5) and discussion (page 14). As a consequence of these results, when referring to HTRA3 we removed “mitochondrial” from the corresponding subtitles in the text, since this protein is not prevalently mitochondrial, at least in IMR-90 cells.

In supplementary Figure 2f it is unclear how 100% of cellular ATP content in PN17 cells is glycolysis-derived while another 100% of cellular ATP content in PN17 cells is derived via oxidative phosphorylation.

Response. In Supplementary Figure 2f of the previous version, values of ATP from glycolysis and ATP from OXPHOS were each expressed as percent of the respective values at PN'17 (100% for glycolysis and 100% for OXPHOS). This representation better showed the glycolytic shift at PN'28, with a large percentual drop of OXPHOS and a considerable increase of glycolysis. Given the comment of the Referee, we worried that this representation may not be clear for a reader, and in the revised version we present these data with cumulated percent of ATP from glycolysis and OXPHOS (see replaced Supplementary Fig 3f).

In supplementary Figure 2 g the decrease of expression of complex I and II is hard to see.

Response. We have raised the contrast specifically in the portion of the blot that displays representative factors of complexes I, II, and IV, and showed this part of the blot separately on the right of panel g. We clearly see a reduction of complexes I and IV at PN'26 and PN'28, but the situation of complex II is not better clarified. Normalization to GAPDH levels showed 15% reduction of complex II at PN'28, which we realize is not robust enough for our claim. We thus replaced the sentence at page 6 from “decreased levels of mitochondrial complexes I-IV” to “decreased levels of mitochondrial complexes I, III, and IV”

At page 6 in line 22 the authors mention that CSA protein is not decreased during senescence but recently it was published that CSA protects also from cellular senescence (Cordisco et al 2019). The authors should comment on this.

Response. As discussed specifically above, in addition to transcripts, CSA protein levels have also been tested now (new lower panel in Fig. 1j), which do decrease with senescence, although not as extensively and as early as CSB, and this point is now discussed in the manuscript. Our data are thus in agreement with experiments of Cordisco et al, performed on keratinocytes from Cockayne syndrome patients *versus* healthy controls.

At page 7 in line 17 the authors should correct the mistake ‘precious’ senescence

Response. We have corrected this typing error.

Reviewer #2 (Remarks to the Author):

Chrochemore et al report that depletion of CSB due to H3 hypoacetylation causes upregulation of p21 and senescence. They also suggest that CSB depletion can promote mitochondrial impairments via overexpression of the HTRA3 proteases. The manuscript is technically ok, but many important controls are missing. Organization of figures and panels is sometime confusing, and there are several overinterpretation of data. While the role of CSB in promoting senescence is quite well demonstrated, the impairment of mitochondrial functions is definitely not evident.

Response. We thank the referee for the comment on the demonstration of the role of CSB in promoting senescence. We have addresses his/her questions below.

Major points:

Fig 1C. What do the authors mean with 'independent of cell size'? Is this normalized somehow? If so, quantification of cell volume should be provided

Response. Since during senescence cells increase in size, the increase of HTRA3 signal that we observe in senescent cells could be simply due to increase in cell surface. However, this is not the case since we measured the mean fluorescence intensity, which is the sum of all pixel intensities divided by the total number of pixels, a method that compares the concentration of a probe in cells independently of the cell surface. We had indicated this procedure and its relevance for comparing signals in cells of potentially different size (lines 4-8 at page 22 of the revised version). We had also included the measurement of cell size during senescence in Supplementary Figure 1b.

Organization of panels in figure 1 is highly confusing

Response. Panel c, which includes an upper and a lower image (for representative images and quantification, respectively) could be confusing. We have now enclosed the upper and lower part of this panel in a frame. Given that all other panels follow the alphabetic order in the regular left-to-right and top-to-bottom order, this figure should be no longer confusing.

Quantification of global and mitochondrial ROS suggests that the difference between pre- and senescent cells is minimal at best. First, images of fluorescence intensity should be provided. Second, more sensitive methods to measure ROS production/accumulation should be utilized

Response. We provide now images of fluorescence intensity for Mitosox (Supplementary Fig. 3e, lower panel), and also of p21 (Supplementary Fig. 3c, right panel) that were missing, and show quantification with more informative scatter plots rather than histograms. We acknowledge that fluorescence methods based on DCF-DA perhaps may not be sensitive enough to detect ROS in our system. Thus, to assess ROS levels we replaced the DCF-DA fluorescence method (former Supplementary Fig. 2d) with a more sensitive method, namely measuring reduced glutathione (GSH), a strong scavenger of ROS, and its ratio with oxidized glutathione (GSSG) using a luminescence assay. We observed a decrease of the GSH/GSSG ratio at PN'26 and PN'28 compared to PN'17-PN'22, suggesting increased oxidative stress in pre-senescent and senescent cells, and confirming our findings. This result is shown in Supplementary Figure 3d, and presented at page 6. This assay reveals differences in oxidative stress that were not detected with the DCF-DA assay. We thank the referee for this request of a more sensitive method to detect oxidative stress.

Figure 2e only suggests a slower population doubling time in shCSB cells. Also, no correlation between this figure and 2f and 2g, where senescence seems to happen much earlier. Authors should try to understand this discrepancy

Response. Fig. 2e (population doubling), Fig. 2f (percent of SA- β -Gal cells), and Fig. 2g (RT-qPCR of $p21^{Waf1}$ and *HTRA3*) belong to the same experiment. We apologize if this was not clear enough in our text.

Thus, CSB depletion, which was demonstrated in the first two passages (PN19 and PN20, see Fig. 2c), resulted in increased percentage of SA- β -Gal⁺ cells (Fig. 2f), increased $p21^{Waf1}$ and *HTRA3* expression (Fig. 2g), and reduced population doubling (Fig. 2e). Moreover, during the next passages (PN21-PN24/PN26, depending on the shCSB) the levels of SA- β -Gal⁺ cells were constantly higher, and the population doubling reduced, in CSB-silenced cells than in shSCR. Please note that cells stopped replicating at PN28 for shSCR, whereas at PN26 for

shCSB#1 and PN24 for shCSB#2. To underscore the reduced population doubling upon CSB silencing, we now indicate significant PNs in CSB-silenced cells in Fig. 2e.

To note also that increased senescence on the long time (several passages after transduction) was a consequence of CSB downregulation during just the first 2-3 passages (PN19-PN21). Indeed, CSB silencing was no longer maintained at later passages (starting at PN21 for shCSB#1, and PN22 for shCSB#2, Supplementary Fig. 4a). These data underscore the pro-senescence effect of transient CSB silencing.

It is not clear what is the phenotypical effect of using anacardic acid on proliferating cells

Response. In previous Fig. 3h we showed reduced H3 acetylation and reduced CSB levels with increasing doses of anacardic acid (AA) for 24h. We performed WB again (using frozen cell pellets from the previous experiment) assessing a larger number of factors, and also upon 24h recovery from AA treatment, which was not shown in the previous version of the manuscript.

These data replace old Fig. 3h and confirm CSB depletion upon reduced H3 acetylation triggered by increasing AA concentrations. To note, in the previous blot migration of the specific CSB and the unspecific band were not resolved due to electrophoresis in 1.5 mm thick gel, whereas in the present blot (as well as the other blots in the paper) electrophoresis was done in 1 mm thick gels (both NuPAGE gels, Invitrogen). CSB depletion in Fig. 3h is compatible with reduced but not depleted CSB transcripts (Fig. 3g). Under these conditions we did not observe immediate increase of p21 and HTRA3 expression, in agreement with their increase a few passages later than CSB depletion (see Fig. 1). Accordingly, 24h after recovery we observed p21 increase from 8 μ M AA treatment, and HTRA3 increase from all concentration of AA, but in particular at 4 μ M. Upon recovery from AA treatment, H3 acetylation levels as well as CSB levels were restored as in untreated controls or vehicle. These data indicate that AA at the highest doses induces senescence and HTRA3 expression following H3 hypocetylation-dependent CSB depletion, and this process is observed 24h after treatment, compatibly with delayed p21 depletion and HTRA3 overexpression upon CSB depletion.

I would expect a more pronounced proliferation in cells without p53, which would be a good control for silencing. However, this doesn't seem to happen. Authors should discuss why

Response. We had shown almost complete depletion of p53 expression upon transduction with shp53 (Fig. 4c), which does demonstrate successful silencing of this gene. As the referee says, one may expect more pronounced proliferation in cells depleted for p53 because of the role of this protein in orchestrating the response to DNA damage, including cell cycle arrest. However, at these early passages in culture it is possible that the DNA damage was not large enough to trigger the p53 response, and thereby increase the proliferation rate of cells depleted of p53. We have discussed this point at page 15.

We noticed that in Fig. 4d at each passage (PN22 and PN23) shp53 cells (light blue line) reached confluence earlier (and were thereby passed earlier) than the p53-proficient control cells (shSCR, dark blue line). As a consequence, the slope of the growth curve appears steeper for shp53, but differences were not significant with the two-way ANOVA test, perhaps due to the limited number of PNs tested. It is possible that at later passage numbers, also expectedly in the presence of more extensive DNA damage, this difference becomes significant, supporting the notion that p53 depletion promotes pronounced proliferation. However, since the scope of this paper was not to test p53-proficient versus p53-depleted proliferation, we did not test further the growth rate of these cells. We nevertheless took into consideration this point of discussion and replaced the sentence at page 11 from “*Notably, p53 silencing alone did not decrease population doubling compared to control shSCR (Fig. 4d)*” to “*P53 silencing alone had little or no effect on the population doubling compared to control shSCR, (Fig. 4d)*”, and added a brief paragraph at lines 9-11 of page 15.

The effect of PFTa on p21 is not shown. Authors should provide evidences that the inhibitor is effective in these experiments

Response. We assessed the levels of p21, which is activated by p53, on shCSB and control shSCR cells treated or not with PFT-a. We observed that PFT-a reduced p21^{Waf1} expression when this locus was upregulated (*i.e.* upon CSB silencing), whereas it has no effect under basal expression (control SCR cells), Fig. 4g. This was also the case at the protein level, Fig. 4h. These experiments confirm that PFT-a inhibits p53-mediated p21 activation in our experiments, as reported at page 11. These experiments also confirm our finding that CSB-mediated senescence is essentially independent of p53 since proficient p53 activity does not promote senescence.

10 days after IR, p16 is normally upregulated in IMR90 cells (but not at earlier time points). It is surprising authors do not find elevated expression

Response. In a previous report, IMR-90 cells displayed increased p16 levels seven days post 10 Gy IR (Noren Hooten and Evans, 2017). Since our experiments were done ten days post 10 Gy IR, we wondered whether the different time point was responsible for this difference in the levels of p16 in the two experiments. We thus assessed p16 levels at an earlier time point (7days post-IR) as well as at 10 days and 15 days post-IR. We did not observe p16 increase in any of these conditions (see Supplementary Fig. 6b).

However, we noticed that in our experiments the level of p16 in untreated cells was possibly higher than in Noren Hootens and Evan's experiment. The higher levels of p16 in non-irradiated controls may explain the lack of further increase upon IR in our experiments. However, it is difficult to compare the absolute levels of a protein in WB done in different labs, with not identical cell culture conditions, and possibly different antibodies (in Noren Hooten and Evans the Ab was not specified). Another difference with the published paper is that they irradiated "young" IMR-90 at 30 passage doublings, whereas in our experiments "young" IMR-90 were at PN16 (at PN30 they are already senescent). Since p16 expression is often induced late after senescence induction and the senescent phenotype is dynamic (Hernandez-Segura et al., 2017) it is possible that p16 is activated differently in these sets of experiments. This point has been discussed at page 12 of the revised manuscript.

Why p21 would go up independently of p53 and CSB upon palbociclib treatment? The conclusion of the figure is definitely not clear

Response. It has been shown that upon palbociclib treatment p21 accumulates due to the release of myc-dependent repression of p21 (Bonelli et al., 2017). Our data are thus in agreement with the literature. This upregulation of p21 is, however, not dependent on CSB. We have inserted a sentence on this point in the discussion (pages 15-16).

Figure 6 is dispensable for the message of the paper, just confusing

Response. Figure 6 shows that the same treatment that rescued HTRA3/POLG1/mitochondrial defects in CS cells (MnTBAP, Chatre et al., 2015) is also to some extent effective in senescent cells, reinforcing the link between the underlying defective processes. Despite CS cells are mutated/depleted in CSB (or CSA), whereas CSB is proficient (but depleted) during replicative senescence, the downstream defects are mechanistically similar. Thus, MnTBAP that acts in response to increased HTRA3, which is due to either CSB impairment (CS, disease) or CSB depletion (senescence in normal cells) shows a possible pharmacological intervention in both cases, and when the correct CSB function/levels cannot be restored. Therefore we believe that these data mechanistically reinforce the finding described in this paper, and we would like to ask to maintain this figure.

References

- Beleford, D., Rattan, R., Chien, J., and Shridhar, V. (2010). High temperature requirement A3 (HtrA3) promotes etoposide- and cisplatin-induced cytotoxicity in lung cancer cell lines. *J Biol Chem* 285, 12011-12027.
- Bonelli, M.A., Digiacomio, G., Fumarola, C., Alfieri, R., Quaini, F., Falco, A., Madeddu, D., La Monica, S., Cretella, D., Ravelli, A., et al. (2017). Combined Inhibition of CDK4/6 and PI3K/AKT/mTOR Pathways Induces a Synergistic Anti-Tumor Effect in Malignant Pleural Mesothelioma Cells. *Neoplasia* 19, 637-648.
- Caputo, M., Frontini, M., Velez-Cruz, R., Nicolai, S., Pranter, G., and Proietti-De-Santis, L. (2013). The CSB repair factor is overexpressed in cancer cells, increases apoptotic resistance, and promotes tumor growth. *DNA Repair (Amst)* 12, 293-299.
- Chatre, L., Biard, D.S., Sarasin, A., and Ricchetti, M. (2015). Reversal of mitochondrial defects with CSB-dependent serine protease inhibitors in patient cells of the progeroid Cockayne syndrome. *Proceedings of the National Academy of Sciences of the United States of America* 112, E2910-2919.
- Cordisco, S., Tinaburri, L., Teson, M., Orioli, D., Cardin, R., Degan, P., Stefanini, M., Zambruno, G., Guerra, L., and Dellambra, E. (2019). Cockayne Syndrome Type A Protein Protects Primary Human Keratinocytes from Senescence. *The Journal of investigative dermatology* 139, 38-50.
- Dekker, F.J., van den Bosch, T., and Martin, N.I. (2014). Small molecule inhibitors of histone acetyltransferases and deacetylases are potential drugs for inflammatory diseases. *Drug discovery today* 19, 654-660.

- Hernandez-Segura, A., de Jong, T.V., Melov, S., Guryev, V., Campisi, J., and Demaria, M. (2017). Unmasking Transcriptional Heterogeneity in Senescent Cells. *Current biology* : CB 27, 2652-2660 e2654.
- Horibata, K., Iwamoto, Y., Kuraoka, I., Jaspers, N.G., Kurimasa, A., Oshimura, M., Ichihashi, M., and Tanaka, K. (2004). Complete absence of Cockayne syndrome group B gene product gives rise to UV-sensitive syndrome but not Cockayne syndrome. *Proceedings of the National Academy of Sciences of the United States of America* 101, 15410-15415.
- Noren Hooten, N., and Evans, M.K. (2017). Techniques to Induce and Quantify Cellular Senescence. *Journal of visualized experiments* : JoVE.
- Selzer, R.R., Nyaga, S., Tuo, J., May, A., Muftuoglu, M., Christiansen, M., Citterio, E., Brosh, R.M., Jr., and Bohr, V.A. (2002). Differential requirement for the ATPase domain of the Cockayne syndrome group B gene in the processing of UV-induced DNA damage and 8-oxoguanine lesions in human cells. *Nucleic acids research* 30, 782-793.
- Wang, Y., Li, F., Zhang, G., Kang, L., and Guan, H. (2016). Ultraviolet-B induces ERCC6 repression in lens epithelium cells of age-related nuclear cataract through coordinated DNA hypermethylation and histone deacetylation. *Clin Epigenetics* 8, 62.
- Yosef, R., Pilpel, N., Papsimadov, N., Gal, H., Ovadya, Y., Vadai, E., Miller, S., Porat, Z., Ben-Dor, S., and Krizhanovsky, V. (2017). p21 maintains senescent cell viability under persistent DNA damage response by restraining JNK and caspase signaling. *The EMBO journal* 36, 2280-2295.

REVIEWERS' COMMENTS:

Reviewer #1 (Remarks to the Author):

All points have been addressed by the authors in a sufficient way.

Reviewer #2 (Remarks to the Author):

I would like to thank the authors for such a complete and detailed revision of the manuscript. My only main concern regards Figure 6, as I still think that its message doesn't add to the paper. As a minor point, I would encourage the authors to try a better organization of the panel 1c

REVIEWERS' COMMENTS:

Reviewer #1 (Remarks to the Author):

All points have been addressed by the authors in a sufficient way.

Reviewer #2 (Remarks to the Author):

I would like to thank the authors for such a complete and detailed revision of the manuscript. My only main concern regards Figure 6, as I still think that its message doesn't add to the paper. As a minor point, I would encourage the authors to try a better organization of the panel 1c

Response: We moved the data from Figure 6 to Supplementary Information (i.e. Supplementary Figure 10), as also suggested by the Editor.

We have also reorganized panel 1c, as well as all other multiple panels in the figures, following the referees concern and the editorial requirement of Nature Communications.

We thank the Referees for their comments that helped clarifying and consolidating our findings.

Please note that after the first revision, several figures and figure panels have been renamed to respond to the last requests of the referees and the editors, and to fulfil the publisher requirements. As a consequence, the figures named in this document do not necessarily correspond to the final figure number/figure panel that appear in the published article.

The changes are indicated in the scheme below.

Revision 1	Revision 2
Figure 1	Figure 1 (contains Suppl Fig 1c) Figure 2
Figure 2	Figure 3 Figure 4
Figure 3	Figure 5
Figure 4	Figure 6
Figure 5	Figure 7
Figure 6	Supplementary Figure 10
Figure 7	Figure 8
Supplementary Figure 1	Supplementary Figure 1
Supplementary Figure 2	Supplementary Figure 2 Supplementary Figure 3
Supplementary Figure 3	Supplementary Figure 4
Supplementary Figure 4	Supplementary Figure 5 Supplementary Figure 6
Supplementary Figure 5	Supplementary Figure 7 Supplementary Figure 8
Supplementary Figure 6	Supplementary Figure 9
Supplementary Figure 7	Supplementary Figure 10
Supplementary Table 1	Supplementary Table 1
Supplementary Table 2	Removed (now in Source Data file)